# The Avalanche Terrain Exposure Scale (ATES) v.2

Grant Statham [1, 2], Cam Campbell [3]

[1] Parks Canada Agency, Banff, AB, Canada
[2] Simon Fraser University, Burnaby, BC, Canada
[3] Alpine Solutions Avalanche Services, Vancouver, BC, Canada

**Correspondence**: Grant Statham (grant.statham@pc.gc.ca)

**Abstract.** The Avalanche Terrain Exposure Scale (ATES) is a classification system that communicates avalanche terrain severity to different target audiences. ATES is a static terrain rating method that is independent of avalanche hazard, so the ratings do not change with the weather and snow conditions. The system was originally introduced in Canada in 2004 as a risk
management tool for public avalanche safety programs and uses two synonymous methods: one for terrain assessment and another for public communication. The ATES method applies technical specifications for assessing avalanche terrain to determine ratings, and it is paired with communication models to convey those terrain ratings to different user groups. ATES ratings are found in guidebooks and route descriptions or displayed spatially as zones on a map, and have been widely applied to public safety programs and workplace avalanche safety plans. This paper introduces ATES v.2, a revised and updated system
that merges the two previous ATES models into a single method that: 1) expands the original version from three levels to five by including Class 0 – Non-avalanche terrain, and Class 4 – Extreme terrain, 2) removes glaciation as an input parameter, and 3) introduces a Communication Model for waterfall ice climbing. The ATES technical specifications are reviewed in detail, along with guidance on their application by field-based practitioners and desktop-based Geographic Information System (GIS) users. The use of both manual and automated ATES assessment methods is discussed, along with methods for presenting
ATES ratings to the target audience. This paper addresses a gap in the literature with respect to avalanche terrain classification for backcountry travel. After twenty years of use in different jurisdictions and countries, the ATES method has not yet been published in a peer-reviewed journal. This publication seeks to correct that and establish a baseline reference for ATES, upon which future terrain-based products and research can build.

## 1    Introduction

The exposure of something vulnerable to avalanche hazard is the definition of avalanche risk (Statham 2008, CAA 2016) and one of the most basic, but important concepts in avalanche risk management; when nothing is exposed, nothing is at risk. Yet most winter backcountry travel scenarios are not this simple, especially with recreational and workplace activities where the elements-at-risk such as skiers, climbers, snowmobilers, or workers are mobile and free to travel unrestricted through the landscape. In these cases, people will encounter terrain choices with different degrees of exposure to avalanche hazard. Their

risk depends upon their route selection and the degree to which they expose themselves to the avalanche hazard , along with their vulnerability to the impacts of an avalanche.

The Avalanche Terrain Exposure Scale (ATES) is an avalanche terrain rating system used to assess and communicate the degree of avalanche terrain exposure. It was first introduced as a guidebook style, terrain rating system for recreational backcountry trips (Statham et al., 2006), then later expanded into a Zoning Model (Campbell and Gould, 2013) to accommodate

spatial applications. Ratings are determined using both subjective and objective criteria and result in a measure of avalanche terrain exposure on an ordinal scale. Unlike the dynamic nature of avalanche hazard assessments, which rise and fall with the changing weather and snowpack conditions, ATES ratings are based upon constant parameters that do not change (e.g., slope angle, exposure) or change slowly (e.g., long-term avalanche frequency, forest density), resulting in a static, unchanging terrain rating.

Since its introduction in Canada in 2004, ATES has been applied in many different jurisdictions and countries (e.g., Mcmanamy et al., 2008; Bogie and Davies, 2010; Gavaldà et al., 2013; Maartensson et al., 2013; Pielmeier et al., 2014; Larsen at al., 2020), has become a widely used risk management and avalanche education tool (Haegeli et al., 2006; Floyer and Robine, 2018; Zacharias, 2020), and has been used as a research tool to measure terrain use preferences (e.g., Sykes et al., 2020; Johnson & Hendrikx, 2021; Hendrikx et al., 2022). In Canada, use of ATES has grown beyond recreational applications into

policy and regulatory frameworks (Parks Canada, 2005b), and is now widely used in workplace avalanche safety plans. Each of these ATES applications has used different approaches to meet different objectives, or to utilize emerging technology . Examples of different techniques include the manual rating of backcountry touring routes (Parks Canada 2004; Baldwin, 2009; Scott and Klassen, 2011; Statham and Hueniken, 2023; Beacon Guidebooks, 2024), mixed GIS and manual mapping/rating of backcountry zones (Gavaldà et al., 2013; Avalanche Canada, 2024) and automated, algorithm-based mapping/rating (Alberta

Parks, 2024; Sykes et al., 2024; Toft et al., 2024). Typically, a more objective approach leads to smaller scale zoning around measurable terrain features, such as in Alberta Parks (2024) and this is different than a manual approach, where terrain is often grouped into zones that are logical for a recreational application (e.g.: Avalanche Canada, 2024), but this requires local expertise. Striking the right balance of objective measurements (e.g.: slope angle), subjective estimates (e.g.: frequency-magnitude) and local knowledge (e.g.: route options) is a challenge for the assessment of any ATES rating. Ultimately, ATES

is a communication tool, and the resulting product must make sense and be easily understood by the receiver of the information. Over the past two decades, advances in technology and geospatial tools have facilitated a broader application of the ATES concept, including automated ATES ratings (Larsen et al., 2020), which greatly expands the potential scope of terrain classification. At the same time, the continued growth of backcountry recreation has furthered the need for improved avalanche terrain tools (Klassen, 2012) to meet the needs of both experienced backcountry users, and people with no appetite for

avalanche risk. Backcountry terrain use patterns have changed, and ATES needs to change with them.

The objectives of this paper are: 1) To introduce an updated version of ATES, now called ATES v.2 , 2) to establish a baseline reference for the ATES methodology in a peer reviewed journal, and 3) to fill a gap in the literature with respect to avalanche terrain classification schemes. We start with an overview and background on avalanche terrain rating systems, followed by a

description ATES v.2, starting with changes from previous versions and then introducing three revised ATES models for assessment and communication. The application of ATES is then described, including methods for the assessment and presentation of terrain ratings followed by discussion on the limitations of the ATES system.

## 2    Background

Terrain rating systems play an essential risk management function in recreational activities such as climbing, hiking, kayaking, skiing and mountain biking. The primary objective of these systems is to simplify complex terrain attributes into easily understood categories that recreationists can use to: 1) understand the difficulty, or severity of their route beforehand to gauge this against their own skills and current conditions, 2) identify and study the crux points of their route ahead of time, and 3) recognize their position on a map in relation to the severity of the terrain around them.

In Canada, avalanche terrain classification systems are either *impact-based* or *exposure-based* (CAA 2016). Traditional hazard mapping methods for land-use planning use impact-based hazard maps (e.g., Rudolf-Miklau et al. 2014; Jamieson and Gould, 2018; Bründl and Margreth 2021), where the frequency and magnitude of avalanches to known locations can be quantified. Hazard maps delineate zones to evaluate and manage risk to infrastructure, roads and occupied structures and can be applied to any asset with a fixed location. However, traditional methods become impractical when the element-at-risk is mobile with unrestricted movement in the landscape, as is the case with backcountry travel. When the element-at-risk can move anywhere, impact-based methods using avalanche frequency and magnitude at fixed locations become impractical because the location of the element-at-risk is constantly changing. Thus, avalanche terrain classification for backcountry recreation requires an exposure-based approach.

Canadian Mountain Holidays (1993) was the first to introduce a static, exposure-based terrain rating system for backcountry skiing, using three terrain categories (A, B and C) and applying these to their inventory of helicopter ski runs. Penniman and Boisselle (1996) proposed a five-level Avalanche Terrain Risk scale based upon terrain severity and modelled after river ratings, which describe the level of difficulty and the consequences of a rapid (Walbridge and Singleton, 2005). Parks Canada introduced ATES v.1/04 (Statham et al., 2006) and rated 275 backcountry ski trips (Parks Canada, 2004) and 75 waterfall ice climbs (Parks Canada, 2005a) in the national parks. Their objective was to encourage guidebook authors to adopt ATES ratings as an aid to the route descriptions in their publications. This method assigned a single rating for each trail, climb or backcountry ski area, and that rating defaulted to the highest terrain class along the entire route or area (Parks Canada, 2004). This method of rating routes has since been described as ATES$_{linear}$ (e.g., Thumlert and Haegeli, 2018).

While ATES$_{linear}$ was effective as a trip planning tool, the application of a single ATES rating for a large area limited its utility for field-based decision making, and for activities unbounded by specific routes, such as snowmobiling. As well, the absence of Class 0 was a notable limitation of ATES v.1/04, because most of the population and most workplaces wish to completely avoid avalanche risk. The ATES Zoning Model (Campbell and Gould, 2013) decoupled ATES from specific routes where the exposure is known, and applied the ratings spatially, as zones on a map. This encouraged a wider adoption of the ATES concept

using an accessible methodology with a reduced and more deterministic set of criteria that was better suited for a GIS environment.

The Zoning Model also introduced and optional Class 0 (non-avalanche terrain) rating, showing where avalanches with consequence are not expected to occur. Avalanche Canada subsequently mapped over 5000 km² of winter backcountry
recreation areas in British Columbia using the Zoning Model (Avalanche Canada, 2024), which has since been described as ATES$_{spatial}$ (e.g., Thumlert and Haegeli, 2018).

Dynamic avalanche risk maps for public recreation were first introduced by the website www.skitourenguru.ch using an algorithm that combined basic terrain characteristics with data from the Swiss avalanche bulletin (Schmudlach and Köhler, 2016a). At the same time, the authors proposed a method for automated avalanche terrain classification (Schmudlach and
Köhler, 2016b) designed to remove the subjectivity in ATES. Following this, Harvey et al. (2018) achieved a major breakthrough using high resolution digital elevation model (DEM) data to combine avalanche terrain characteristics with the avalanche simulation model RAMMS::EXTENDED, and produce avalanche terrain maps for all of Switzerland. Their method was later refined to better communicate the resulting terrain classifications, and incorporate the ATES system (Harvey et al., 2024).

To produce avalanche terrain maps and ratings at a national scale, automated models must be used (Bühler et al., 2018), and it was obvious that the efficiency of automated methods far exceeded that of manual mapping, which is time and labour intensive. To that end, AutoATES, an automated method of applying ATES ratings, was developed in Norway to create nationwide avalanche terrain maps (Larsen et al., 2020), and later updated to AutoATES v.2 (Toft et al., 2024), which aligns with the ATES v.2 described herein.

**3    Primary changes to the Avalanche Terrain Exposure Scale**

Despite sharing the same name, there are significant differences between ATES v1/04 Technical Model and the ATES Zoning Model that have been corrected in ATES v.2. ATES v1/04 was designed to be subjective, applied to recreational routes in the style of a guidebook (backcountry travel routes and waterfall ice climbs), and typically resulted in a single terrain rating that defaulted to the highest ATES class on that route. The Zoning Model aimed to be more objective and GIS-based by introducing
thresholds for slope angle and forest density to encourage smaller scale, spatial applications which included Class 0, but did not consider key parameters such as exposure, avalanche frequency and route options. Both models had strengths and weaknesses and it was clear that an updated ATES v.2 could accommodate both the objective parameters from the Zoning Model and the subjective parameters from ATES v1/04 Technical Model, brought together into a single system utilizing the best parts of both models.

Accordingly, the original ATES v.1/04 Technical Model and the ATES Zoning Model are now merged into ATES v.2, and the ratings have been expanded from three to five levels of terrain exposure. This reflects important backcountry use patterns on both ends of the risk spectrum: from conservative, no-risk Class 0 – Non-avalanche terrain to more aggressive, high-risk

Class 4 – Extreme terrain. Additionally, glaciation has been removed as in input parameter to ATES v.2, and ATES for Waterfall Ice Climbing is introduced as a Communication Model for that activity.

**3.1    Class 0 – Non-Avalanche Terrain**

Class 0 was first introduced by Campbell and Gould (2013) and is now being integrated into ATES v.2. Non-avalanche terrain is arguably the most important rating level because explicitly identifying trails and zones where avalanche do not occur is an essential service for the thousands of tourists who visit mountain areas each winter and want to completely avoid avalanche risk. Groups such as youth groups, tourist hikers, industrial camps and workplace safety requirements often demand a complete

avoidance of avalanche risk. To meet this need, land managers require simple ways to direct people towards non-avalanche terrain. Figure 1 illustrates trails that are rated Class 0 in the immediate vicinity of Lake Louise, Canada, where millions of people visit annually and almost all of them seek to completely avoid avalanche risk.

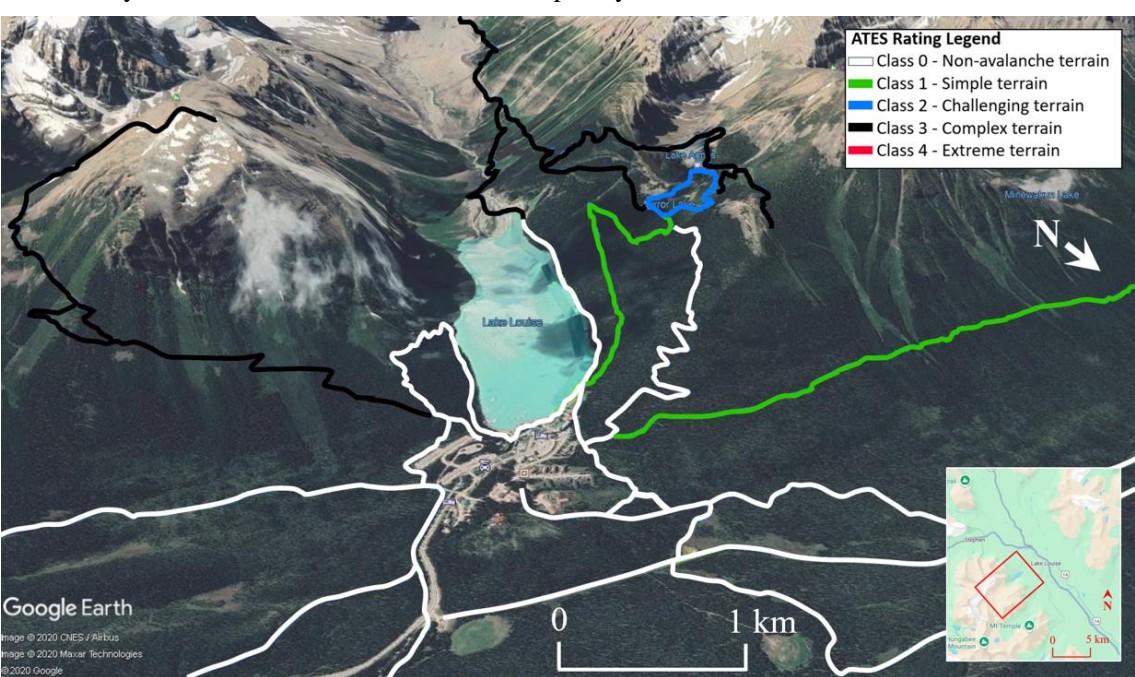

**Figure 1: Designated hiking, snowshoeing and track set cross country ski trails in the Lake Louise area of Canada's Banff**
**National Park with white trails showing the Class 0 – Non-avalanche terrain in the area.**

Although it is a basic competency of an avalanche professional to identify where avalanches can occur and where they cannot, this task is not trivial. For land-use planning applications, determining an avalanche free perimeter is a complex process involving vegetation analysis, mapping of historic events, climate analysis and runout modelling (Jamieson and Gould, 2018). This level-of-effort is usually impractical for mapping backcountry avalanche hazard. Determining a Class 0 – Non-avalanche

Terrain rating requires high confidence in the assessment and can have little to no uncertainty. For this reason, the use of ATES Class 0 is optional, and Class 1 can include Class 0 terrain.

### 3.2    Class 4 – Extreme Terrain

In previous versions of ATES, Complex terrain had a broad criteria that encompassed much of the popular terrain used for alpine recreation, specifically alpine ski touring, snowmobiling and ice climbing. According to backcountry skiing guidebooks
for western Canada, 71% of ski tours in the Coast Range (Baldwin, 2009) and 76% in the Canadian Rocky Mountains (Scott and Klassen, 2011) are rated Class 3 – Complex terrain. Harvey et al. (2018) considered ATES to have limited practical value in the Swiss Alps because too many tours would inherently be classified as Complex. This lack of a finer resolution within Complex terrain has limited the value of an ATES rating for experienced recreationists who spend much of their time in steep mountain terrain. As backcountry recreation continues to grow, this style of terrain is becoming more popular. Freeriding and
ice climbing routinely travel through or below high consequence avalanche terrain that presents as its own distinct class of terrain, now known as Class 4 – Extreme Terrain (Figure 2).

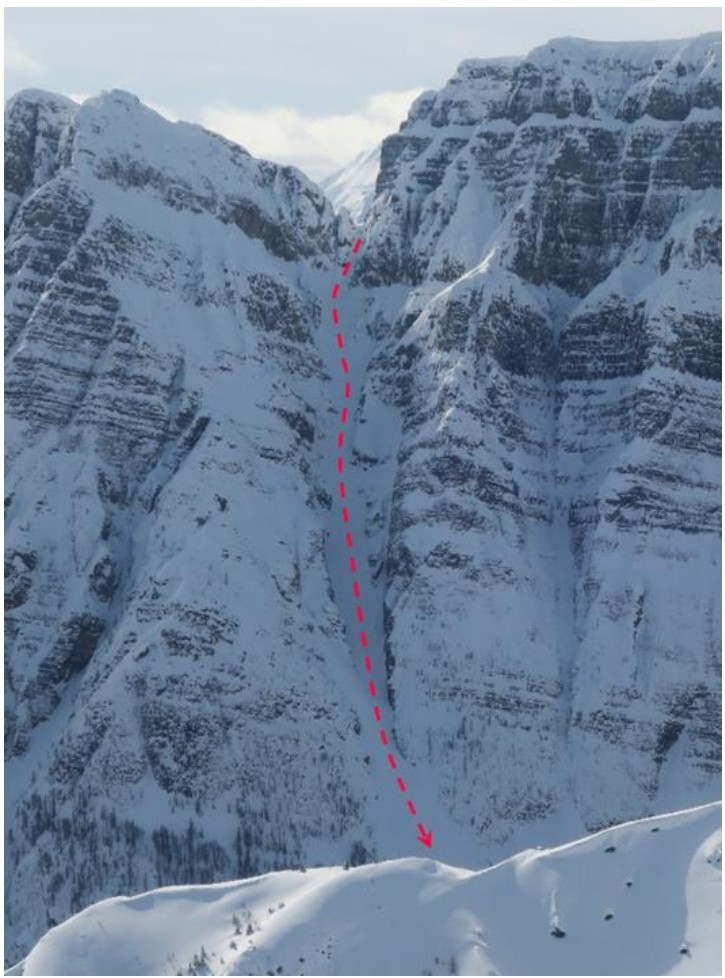

**Figure 2: The Kindergarten Couloir, a popular 1100m freeriding route in Canada's Kootenay National Park rated ATES Class 4 – Extreme terrain due to its sustained exposure (ascent/descent), high slope angle, very high avalanche frequency (>1:1) and no options**
**to reduce exposure. This is a place where and even small avalanches can be fatal.**

### 3.3 Removal of glaciation

Glaciation was an important parameter in the original ATES v.1/04 (Statham et al., 2006), and all glaciated terrain automatically defaulted into at least Class 2 – Challenging terrain, irrespective of any other ATES parameters; there was no Class 1 – Simple Terrain on a glacier. If a glacier presented with "broken or steep sections of crevasses, icefalls or serac exposure", then the rating defaulted to Class 3 – Complex terrain. This was intended to capture the complexity of glacier travel but had the effect of defaulting flat or low angled glaciers into an ATES Class 2 rating, even when there was little or no avalanche terrain. Notably, the ATES Zoning Model (Campbell and Gould, 2013) did not consider glaciation, creating a potential conflict between assessments using these two models.

ATES is primarily concerned with terrain exposed to snow avalanche hazard. Ice avalanches are distinct from snow avalanches in that their failure mechanism follows a different process (Pralong et al., 2005), leading to their inherent unpredictability by field practitioners. For these reasons, glaciation as an independent parameter has been removed from ATES v.2, but crevasses remain as a terrain trap consideration. This will have the effect of down classifying low-angled glaciated terrain that was previously Class 2 – Challenging terrain, into Class 1 or Class 0 terrain.

## 4 Avalanche Terrain Exposure Scale v.2

ATES v.2 is an ordinal, five-level terrain rating system that helps people gauge their exposure to avalanche-prone terrain, and it follows the communication theory of *source-channel-receiver* (Wogalter et al., 1999). The source is the person, or group doing the assessment and determining the rating, the channel is the method of communication (e.g., website, app, guidebook, etc.), and the receiver is the end user of the information.

ATES is a terrain model with dual objectives: assessment and communication. The Communication Models (Tables 1 and 2) are simple by design to achieve the primary objective of ATES by conveying terrain ratings to different receiver groups. The Technical Model (Table 3) is designed for the source (i.e., the terrain assessor) as a specialized reference for identifying, analysing, and classifying avalanche terrain exposure. Although these different ATES models use different language to achieve different objectives, they are synonymous and their thresholds correspond: i.e., ATES says the same thing in two different languages, one technical and one non-technical. The system uses numbers, signal words and colours as options to communicate the rating level.

### 4.1 ATES Communication Models

ATES was born from a Canadian backcountry avalanche disaster where seven high-school students were killed while on an outdoor education school trip in February 2003. Upon review, it became clear that public safety agencies needed better tools to help the public determine what was serious avalanche terrain, and what was not (O'Gorman et al., 2003). Risk communication was the original objective of ATES, and it remains its primary objective today. Regardless of the techniques

used for the assessment of terrain exposure, ATES ratings must ultimately meet the criteria specified in Tables 1 and 2, as these are what is published to the receiver groups.

The Communication Models describe terrain ratings in the language of the receiver group and are light on technical detail with a priority on comprehension. Tables 1 and 2 describe and rank avalanche terrain in a simple way, similar to how the avalanche danger scale (Statham et al., 2010; Avalanche Canada, 2022; EAWS, 2024) describes and ranks avalanche danger; they represent the summary output of a technical assessment, intended for public avalanche risk communication.

When used in combination, models of avalanche danger and models of terrain offer a simplistic, but powerful way to illustrate good risk management through the interaction of snow, terrain and people (Haegeli et al., 2006), and offer a preview into a future where dynamic avalanche risk maps combine these models automatically (e.g.: Schmudlach and Köhler, 2016a).

**Table 1: ATES for backcountry travel.**

| Terrain rating | Class | Description for backcountry travel |
|---|---|---|
| **Non-Avalanche** | **0** | No known exposure to avalanches. Very low-angle or densely forested slopes located well away from avalanche paths, or designated trails/routes with no exposure to avalanches. |
| **Simple** | **1** | Exposure to low-angle or primarily forested terrain. Some forest openings may involve the runout zones of infrequent avalanches and terrain traps may exist. Many options to reduce or eliminate exposure. |
| **Challenging** | **2** | Exposure to well-defined avalanche paths, starting zones, terrain traps or overhead hazard. With careful route finding, options exist to reduce or eliminate exposure. |
| **Complex** | **3** | Exposure to multiple overlapping avalanche paths or large expanses of steep, open terrain. Frequent exposure to overhead hazard. Many avalanche starting zones and terrain traps with minimal options to reduce exposure. |
| **Extreme** | **4** | Exposure to very steep faces with cliffs, spines, couloirs, crevasses or sustained overhead hazard. No options to reduce exposure; even small avalanches can be fatal. |

Waterfall ice climbing is a specialized activity, often very exposed to avalanche hazard and high risk (Statham and Hueniken, 2023). Ice climbers are a unique audience in that their routes are commonly inside avalanche paths, meaning that climbers can be exposed for long periods of time to slopes overhead that cannot be assessed in conventional ways. The primary emphasis of ATES for waterfall ice climbers is exposure time and avalanche frequency. How frequently does the route avalanche, and how long will climbers be exposed to it?

**Table 2: ATES for waterfall ice climbing.**

| Terrain rating | Class | Description for waterfall ice climbing |
|---|---|---|
| Non-Avalanche | 0 | Routes with no exposure to avalanches except small sluffs and spindrift. |
| Simple | 1 | Routes with brief exposure to very low frequency avalanches starting from above or crossing occasional short slopes. |
| Challenging | 2 | Routes with long exposure to low frequency avalanches or brief exposure to high frequency avalanches starting from above or crossing a few short slopes. Options exist to reduce exposure. |
| Complex | 3 | Routes with long exposure to high frequency avalanches starting from above or crossing steep slopes with terrain traps below. Minimal options to reduce exposure. |
| Extreme | 4 | Routes with long and sustained exposure to very high frequency avalanches starting from above and crossing multiple steep slopes with terrain traps below. No options to reduce exposure. |

## 4.2    ATES Technical Model

The ATES Technical Model (Table 3) is designed for avalanche terrain assessment and is used to determine an ATES rating. The model breaks down avalanche terrain exposure using eight different parameters:

1.    Exposure
2.    Slope angle and forest density
3.    Slope shape
4.    Terrain traps
5.    Frequency-magnitude
6.    Starting zone size and density
7.    Runout zone characteristics
8.    Route options

Any given Area, Zone, Corridor or Route usually includes terrain criteria that fit into different ATES rating levels, and combining these into a single rating is a subjective exercise with some guidance provided in the following subsections. Not all eight parameters will be able to be assessed every time, particularly at smaller scales. For example, assessing starting zone size and density implies that there are multiple starting zones, assessing exposure and route options implies that a route has been selected, and assessing slope shape often requires more than one slope to assess. Sometimes certain parameters will simply
not be apply to the assessed terrain. For these reasons, none of these criteria are mandatory, and the assessor must gather and work with the best information available to them.

Within a total of 40 criteria, there are six ***bold defaults*** that when met, automatically default the ATES rating into that category or higher. Otherwise, the overall rating is an evaluation based predominantly on expert judgement that involves: 1) analysing

the terrain against each ATES parameter for best fit, 2) comparing this to levels above and below, and 3) deciding what the best overall ATES rating is. Field checking and peer review of ATES ratings from other qualified individuals is important for error correction, accuracy and ultimately improving confidence in the assessment.

The following sections provide guidance for evaluating each of the eight parameters that define the ATES Technical Model (Table 3) by describing their influence on terrain severity and the range of thresholds from Class 0 to Class 4 .

### 4.2.1 Exposure

Exposure is the situation of people, infrastructure, housing or other tangible assets located in hazard-prone areas (United Nations, 2016). With respect to avalanche risk, exposure is the extent to which an element at risk is subject to avalanche hazards, and is a function of both space and time (CAA, 2016). In other words: *where*, and *for how long* something is subject to an avalanche hazard. Exposure is a crucial ingredient for avalanche risk and without it, there is no risk.

Spatial exposure considers precisely where an element at risk is located in the terrain and their position relative to the surrounding avalanche hazard, including overhead hazard. This is fundamental, because even during periods of high avalanche hazard, a simple reduction in spatial exposure will reduce the risk. On small-scale terrain features, even minor adjustments in how one is exposed to the hazard will change their risk – a few meters in either direction can be the difference between a low and high-risk situation (Statham, 2008). ATES uses the terminology: none, runouts only, single paths, multiple paths and inside/under starting zones to describe the range of spatial exposure.

ATES considers temporal exposure in two different but related ways: the assessment of an ATES rating examines temporal exposure in terms of how long an element-at-risk is exposed. For example, being under an avalanche path for 10 minutes presents a higher severity than being exposed to the same path for only one minute. This kind of temporal exposure applies directly to field techniques used to manage the risk: which is better, taking 10 minutes and crossing under one at a time? Or taking one minute and crossing as one large group of people? The terminology: minimal, brief, intermittent, long, frequent and sustained used in Tables 2 and 3 refers the length of time one should expect to be exposed. The application, or use of ATES ratings as a tool for risk management, asks the receiver to consider temporal exposure in terms of when different classes of terrain are within their risk threshold, and when they are not. This is a dynamic avalanche risk assessment which requires combining the ATES rating (static) with an avalanche hazard assessment (dynamic). For example, when the hazard is Low, then Complex terrain may be appropriate; conversely, when the hazard is High, then Complex terrain may be inappropriate and Simple terrain a better choice.

ATES considers both actual and potential exposure, depending on the approach. ATES$_{linear}$ rates specific, pre-defined Routes, meaning that the actual exposure is known and can be evaluated, whereas ATES$_{spatial}$ rates Areas or Zones of terrain without a specific route, which is potential exposure. Once the receiver of the information plans a specific route, then their actual exposure becomes known, and the ATES ratings can be utilized.

**Table 3: ATES Technical Model.** *Bold italicized text* indicates default values that automatically place the ATES rating into that category or higher.

| | Class 0 Non-Avalanche Terrain* | Class 1 Simple Terrain | Class 2 Challenging Terrain | Class 3 Complex Terrain | Class 4 Extreme Terrain |
|---|---|---|---|---|---|
| **Exposure** | No known exposure to avalanche paths | Minimal exposure crossing low-frequency runout zones or short slopes only | Intermittent exposure managing a single path or paths with separation | *Frequent exposure to starting zones, tracks or multiple overlapping paths* | Sustained exposure within or immediately below starting zones |
| **Slope angle** *and* **Forest density** | Very low angle (< 15°) open terrain unconnected to steeper slopes, or steeper areas in dense forest. | Low angle (15°-25°) open terrain with isolated small (< Size 2) moderate angle slopes and/or forest openings for runout zones. | Moderate angle (25°-35°) open terrain with isolated large (≤ Size 3) high angle slopes in glades or open areas. | Large proportion of high angle (35°-45°) open or gladed terrain, but mostly moderate angle terrain. | Large proportion of very high angle (>45°) terrain with few or no trees. |
| **Slope shape** | Straightforward, flat or undulating terrain | Straightforward undulating terrain | Mostly undulating with isolated slopes of planar, convex or concave shape | Convoluted with multiple open slopes of intricate and varied terrain shapes | Intricate, often cliffy terrain with couloirs, spines and/or overhung by cornices |
| **Terrain traps** | No avalanche-related terrain traps | Occasional creek beds, tree wells or drop-offs | Single slopes above gullies or risk of impact into trees or rocks | Multiple slopes above gullies and/or risk of impact into trees, rocks or crevasses | Steep faces with cliffs, cornices, crevasses and/or risk of impact into trees or rocks |
| **Frequency-magnitude** (avalanches:years) | *Never > Size 1* | < 1:100 - 1:30 for ≥ Size 2 | 1:1 for < Size 2 *1:30 - 1:3 for ≥ Size 2* | 1:1 for < Size 3 *1:1 for ≥ Size 3* | 10:1 for ≤ Size 2 > 1:1 for > Size 2 |
| **Starting zone size and density** | No known starting zones | Runout zones only except for isolated, small starting zones with < Size 2 potential | Isolated starting zones with ≤ Size 3 potential or several start zones with ≤ Size 2 potential | Multiple starting zones capable of producing avalanches of all sizes | Many very large starting zones capable of producing avalanches of all sizes |
| **Runout zone characteristics** | No known runout zones | Clear boundaries, gentle transitions, smooth runouts, no connection to starting zones above | Abrupt transitions, confined runouts, long connection to starting zones above | Multiple converging paths, confined runouts, connected to starting zones above | Steep fans, confined gullies, cliffs, crevasses, starting zones directly overhead |
| **Route options** | Designated trails or low-angle areas with many options | Numerous, terrain allows multiple choices; route often obvious | *A selection of choices of varying exposure; options exist to avoid avalanche paths* | Limited options to reduce exposure; avoidance not possible | No options to reduce exposure |

* The use of Class 0 is optional due to the reliability needed to make this assessment; otherwise, Class 1 includes Class 0 terrain.

## 4.2.2 Slope angle and forest density

Slope angle is the primary terrain factor in avalanche release. Slab avalanches typically initiate within the range of 25-55° (McClung and Schaerer, 2023), with most initiating on slopes that have an incline of 30-45°. Within any single slope, the steepest part of the slope is what matters most. This is known as the "critical slope", which is the steepest angle from the horizontal averaged over 10-20 m in the starting zone. (Schweizer et al., 2003; McClung and Schaerer, 2023). ATES associates common slope angle terminology with a range of slope angle values (Table 4).

**Table 4: ATES slope angle terminology and associated values.**

| Slope angle | Slope angle range |
|---|---|
| Very low angle | < 15° |
| Low angle | 15° - 25° |
| Moderate angle | 25° - 35° |
| High angle | 35° - 45° |
| Very high angle | > 45° |

The relationship between slope angle and avalanche release is modified by forest cover (Figure 3) because dense trees can anchor the snowpack to the slope and reduce or eliminate the avalanche hazard. The degree of anchoring effect depends on tree spacing and stem diameter (Weir, 2002; Rudolf-Miklau et al., 2014) as well as crown coverage and ground roughness from lying or standing trees. Forest cover also modifies the snowpack structure by sheltering the snowpack from wind effects and blocking incoming and outgoing solar radiation. Bebi et al. (2009) describe the physical processes that stabilize the snow cover in the forests and modify the effects of terrain factors to include: (i) interception of falling snow, (ii) modification of the radiation and temperature regimes, (iii) reduction of near surface wind speeds, and (iv) direct support of the snowpack by the stems.

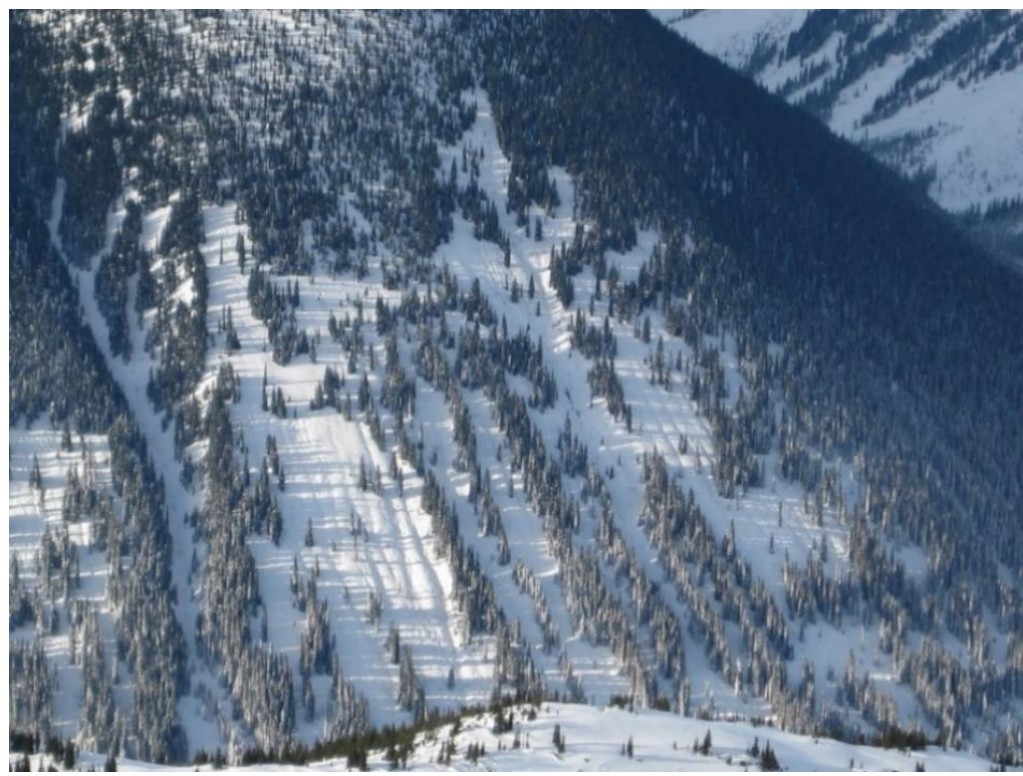

**Figure 3: The interaction between slope angle and forest density is illustrated here where dense forest anchors the snowpack while the steep, open glades are avalanche paths.**

This interaction between forests and avalanches is a complex phenomenon which has been simplified for its application to ATES to examine only tree spacing and its effect on anchoring the snowpack to the slope. Direct support of the snowpack by

tree stems can prevent slab avalanche formation, but primarily in dense forests with more than 1000 stems per hectare (Salm, 1978). In steep forests with less than 1000 stems per hectare, natural and human triggered slab avalanches are common, but minimal research exists on the effects of tree spacing on human triggering of avalanches. Good quality forest cover data is challenging to source, although improving each year. In the absence of good data, ATES uses manual estimates of tree spacing by measuring the typical space between trees and then extrapolating, or averaging across an area. The size of forest openings

can be measured, and Table 5 defines typical spacing for open, gladed and dense forest. Often, significant differences in forest density will delineate the edge of a zone.

**Table 5: ATES forest density terminology and associated values (adapted from Campbell and Gould, 2013).**

| Forest density | Tree spacing[1] | Stem density |
|---|---|---|
| Open | > 10 m average tree spacing | < 100 stems/ha |
| Gladed | 3.2 – 10.0 m average tree spacing | 100 – 1000 stems/ha |
| Dense | < 3.2 m average tree spacing | > 1000 stems/ha |

[1]Based on a minimum stem diameter of 16 cm.

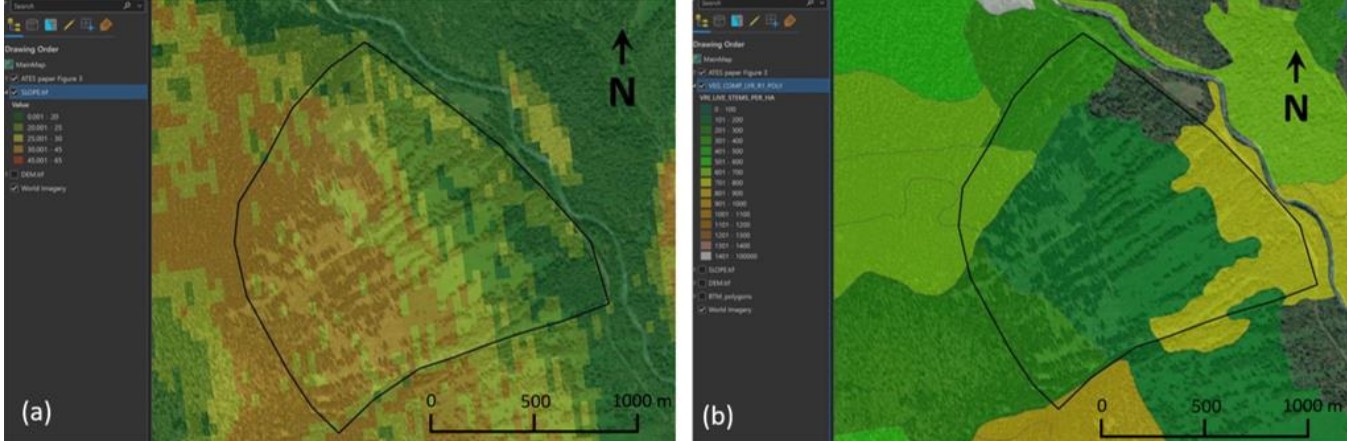

Figure 4: Slope angle distribution (a) across the area shown in Figure 3 indicates the lower half of the area in the 20°- 30° range and the upper half is 30°- 45° while the forest density distribution (b) of the main open/gladed area in the middle is 120 stems/ha, slightly < 10 m spacing. Combined, these thresholds put the overall rating of this area as Class 3 – Complex terrain. Data source: Natural Resources Canada. Basemap source: Esri.

Table 6 shows combined thresholds of slope angle and forest density (Campbell and Gould, 2013), and these proportions can be applied when using GIS tools (Figure 4). For example, the term *large proportions* in Table 3 means > 45% of the terrain (Table 6). From a practical perspective, average tree spacing is done by estimating the distance between individual stems in various locations, and then applying this to the entire slope to get an average value. The largest forest openings in Figure 3 are 760 m long x 170 m wide, with slopes angles of 30°- 45°, so there is little protection from avalanches here. For skiers descending this slope, it would be possible to sneak through this terrain in unstable conditions by following the contiguous strips of dense forest, however these are very close to the large open glades with limited options to reduce exposure.

Table 6: Slope angle and forest density combined thresholds for GIS applications (adapted from Campbell and Gould, 2013).

| Forest Density | 0 - Non-Avalanche | 1 - Simple | 2 - Challenging | 3 - Complex | 4 – Extreme |
|---|---|---|---|---|---|
| Open | $99\% \leq 20°$ | $90\% \leq 20°$ $99\% \leq 25°$ | $90\% \leq 30°$ $99\% \leq 40°$ | $< 20\% \leq 25°$ $45\% > 35°$ | $< 20\% \leq 35°$ $45\% > 45°$ |
| Gladed | $99\% \leq 25°$ | $90\% \leq 25°$ $99\% \leq 35°$ | $90\% \leq 35°$ $99\% \leq 45°$ | | |
| Dense | $99\% \leq 30°$ | $99\% \leq 35°$ | $99\% \leq 45°$ | | |

*Slope angles are averaged over a fall-line distance of 20-30 m.

The overall terrain rating for the area shown in Figure 3 would be Class 3 – Complex terrain, as single ratings usually default to the highest level within the area. However, smaller scale zoning would consider the different distributions of forest density and slope angle, resulting in zones of Class 1 and 2 and 3 terrain (Figure 5).

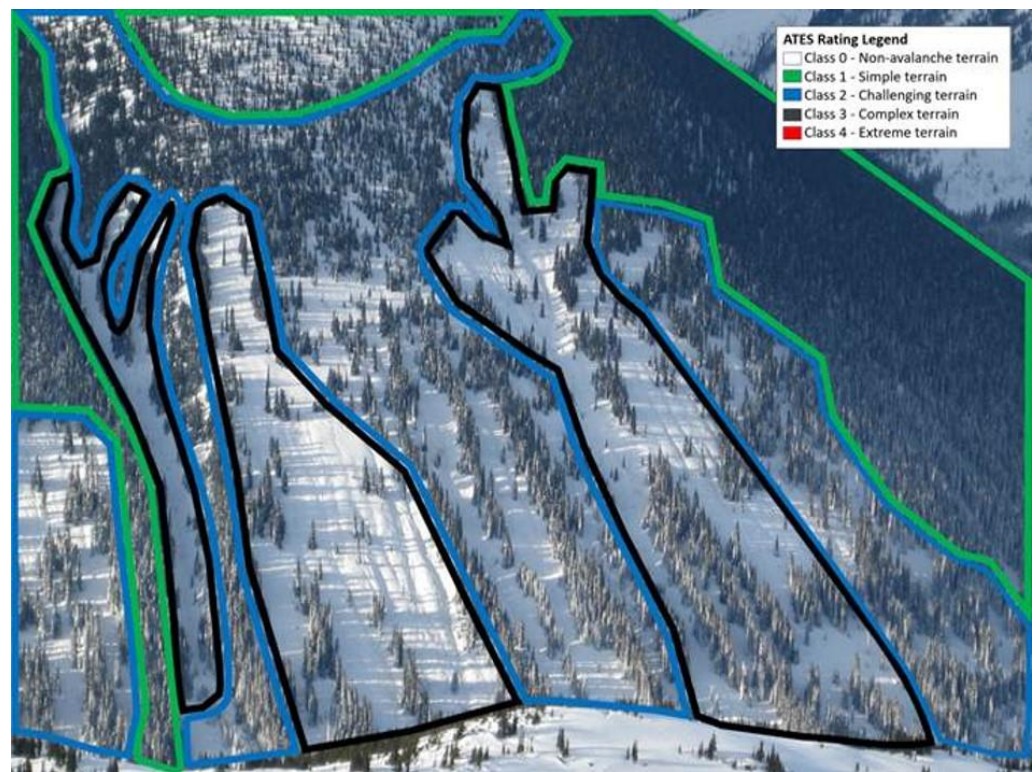

**Figure 5: ATES zoning based on the combination of slope angle and forest density shows zones of Class 1, 2 and 3 terrain across the area in Figure 3.**

### 4.2.3 Slope shape

The shape of snow-covered slopes plays an essential role in route-finding through avalanche prone terrain. During backcountry travel, risk is routinely reduced by carefully weaving through terrain features and relying on their shape to manage spatial exposure. Stopping on high ground to keep people above the flow of an avalanche, using the terrain's shape to set a track that avoids trigger spots and overhead hazard, minimizing spatial and temporal exposure whenever possible, and avoiding steep, unsupported (convex) slopes are all best practices of professional mountain guides (ACMG, 2023). The more convoluted the

slope shape is, the more complicated it is to travel through it.

Although slope curvature is a source of tensile stress (McClung and Schaerer, 2023), the effects of microtopography and slope curvature on avalanche release are not well understood. Convex terrain is said to be unsupported because in the vertical axis, it rolls over at the top of the slope and becomes steepest near the bottom (i.e., the toe of the slope). Convexities add tension to the snowpack and are common trigger points given additional load (Landrø et al., 2020). Even when an avalanche is triggered

from low on the slope, below the convexity, the crack radiates outward from the trigger point, propagating upslope, downslope, and across the slope. The upslope portion of the crack frequently arrests on convexities, where a tensile fracture forms the crown face (Trottet et al., 2022).

Furthermore, conventional avalanche safety has traditionally taught avoidance of convex terrain in favour of planar or concave slopes when route finding (Ferguson and LaChapelle, 2003; Avalanche Canada 2010), because concave slopes are thought to

330 have less tensile stress and better toe-support. In Canadian helicopter skiing, the most frequently closed ski runs (i.e.: most hazardous), are characterized as having more unavoidable, unsupported terrain shapes (Sterchi and Haegeli, 2019). However, recent research into avalanche accidents and the terrain-use patterns of professional guides shows more accidents on planar and concave terrain (Vontobel et al., 2013; Harvey et al., 2018), and that professional guides tend to choose planar terrain in their route selection (Thumlert and Haegeli 2018).

Convoluted terrain also presents more spatially variable snowpack stability compared to planar terrain, because the depth and distribution of the snow is non-uniform. This is primarily due to redeposition from wind effects across uneven topography, both scouring and loading snow around micro terrain features. These wind effects in convoluted terrain increase spatial variability, which is directly related to more trigger points and greater uncertainty in snow slope stability evaluation (Schweizer et al., 2008). As the variance increases, it creates more trigger spots on the slope because it creates more areas where the slab

is thinner and the weak layer can be triggered (Meloche et al., 2024). Zones of terrain that present mixed shapes of concave, convex and planar (Figure 6) usually present a snowpack with more trigger points than zones with a smooth, evenly distributed snowpack where the depth and layer distribution is more predictable.

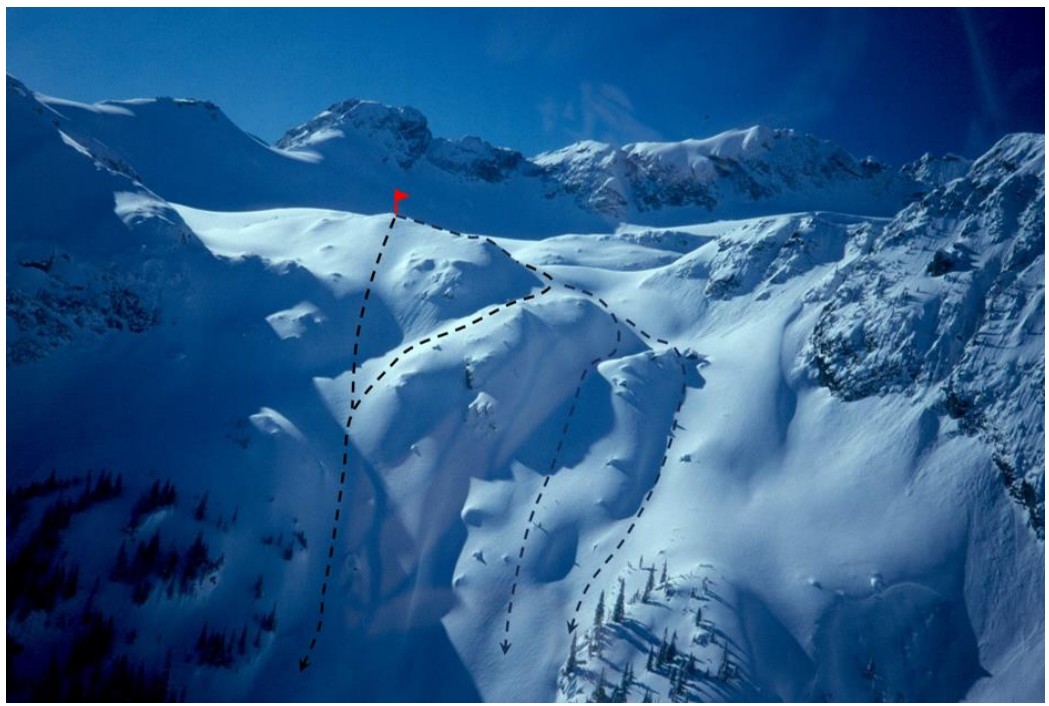

**Figure 6: A helicopter ski run rated Class 3 – Complex terrain, where the slope shapes are convoluted and include gullies, convex**
**rolls, concave slopes and rocky, thin snowpack areas. Dashed lines indicate typical descent routes, weaving around convexities to reach planar, well-supported terrain shapes with more consistent snowpack depth and avoiding obvious trigger points.**

The shape of an individual slope is not usually the defining criteria within ATES; i.e., one convex or concave slope is unlikely to determine the rating, unless that single slope forms the crux of the route. Instead, slope shape should be considered in the aggregate across a larger area, recognizing the influence of that terrain's shape on both avalanche triggering and route finding. Large areas of convoluted terrain are more complex to deal with than large areas of planar terrain, even though in planar terrain there may be fewer options for safe travel. The ATES Technical Model (Table 3) uses the following terms to describe progressively increasing severity in slope shape: flat, undulating, planar, concave, convex, convoluted, intricate and cliffy. Our understanding of the effects of slope shape on avalanche behaviour are not well understood, and not well-supported in the literature. This, despite strongly held convictions by experienced mountain and ski guides, who maintain that the shape of the terrain is one of the most important influences on their route selection. This topic is rich with opportunity for future research.

### 4.2.4    Terrain traps

Terrain traps are topographic features in avalanche paths that increase the consequences of being caught in an avalanche, including serious injury or death from an otherwise harmless avalanche. While the mass of snow in a Size 1 avalanche (Table 8) is not enough to bury a person on a smooth slope, it can be forceful enough to push them off a cliff, or bury them in a gully where the avalanche debris concentrates and becomes locally deep.

Campbell and Gould (2013) categorized terrain traps into those that increase the likelihood and depth of burial, and those that can cause trauma to someone caught in a flowing avalanche. For example, gullies, depressions, and abrupt transitions concentrate avalanche flow, resulting in an increased depth of accumulated debris (Figure 7), while being carried over cliffs or impacting trees, rocks and other downslope obstacles can result in trauma. Trauma has been shown to be the primary cause of death in 20%-30% of avalanche fatalities (Boyd et al., 2008, Sheets et al., 2018, McIntosh et al., 2019). Campbell and Gould (2013) then ranked the severity of terrain traps in terms of increasing consequences from an otherwise harmless avalanche to one that can cause partial burial, minor injury, complete burial, or serious/fatal trauma. Harvey et al. (2018) calculated burial and fall potential using high resolution DEM to create a raster-based layer describing avalanche consequences.

ATES v.2 uses exposure to physical terrain traps such as gullies, cliffs, trees and crevasses as a measure of terrain severity, and an increase in the number and severity of these terrain traps will have a corresponding effect on the ATES rating.

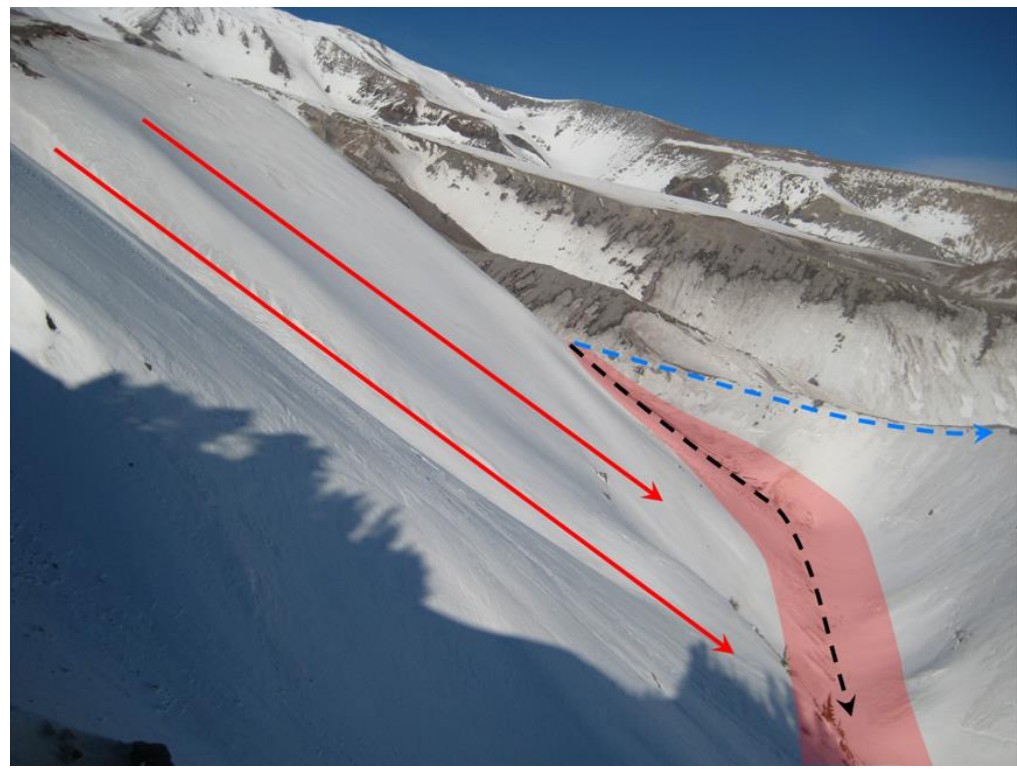

**Figure 7: A dangerous terrain trap where avalanches run down the red flow lines and accumulate deep avalanche debris in the gully below (pink deposition area). The black route is rated Class 3 – Complex terrain and is a poor route choice due to the unavoidable terrain trap, whereas the blue route on the crest of the moraine is Class 2 – Challenging terrain because it avoids most of the exposure.**

### 4.2.5    Frequency and magnitude

The frequency of a natural hazard is the number of times it occurs within a specified time interval (Jackson, 2013). Avalanche frequency within a specific avalanche path is the expected (average) number of avalanches per unit time reaching or exceeding a location (CAA, 2016). This is typically expressed in units of avalanches per year as a ratio that ranges from 1:1 (i.e., one avalanche per year) up to 1:300 (i.e., one avalanche in 300 years). Avalanche paths producing multiple avalanches per year can also be described in the same way (e.g., 3:1 is three avalanches per year).

In practice, formal assessments of avalanche frequency are commonly done during the avalanche planning process for infrastructure developments such as roads or buildings, but this practice is less common for recreation. Avalanche frequency is commonly expressed using terminology such as *low* and *high* which corresponds to a set of frequency ranges (Table 7). Avalanche frequency can be difficult to assess accurately. With good records kept over a long-enough period, reasonable estimates of long-term frequency can be made. But in the absence of good records, avalanche frequency estimates are a subjective exercise using a combination of local knowledge, records, stories, modelling, and indirect observations such as dendrochronology (Carrara, P., 1979). These are often rough estimates, but a lack of formal records does not diminish the importance of avalanche frequency and its influence on avalanche risk assessments and ATES ratings.

For backcountry travel applications, avalanche frequency is a critical measure of terrain severity, i.e., terrain that is known to produce avalanches more frequently is comparatively more dangerous than terrain that produces avalanches less frequently. Commercial backcountry operations are acutely aware of their high-frequency locations and treat them with respect when doing risk assessments. Accordingly, avalanche frequency carries significant weight as an ATES parameter, both in the assessment and communication of the avalanche terrain ratings (Tables 1 and 2). Thresholds for frequency-magnitude are the dominant defaults in the ATES Technical Model (Table 3), meaning that if that threshold is met, then the terrain rating defaults into that category or higher.

Avalanche frequency is the only ATES parameter that considers the influence of the snowpack. This is possible within a static rating system because frequency is a long-term measurement that depends on snow climate (Haegeli and McClung, 2007) rather than short-term weather fluctuations. Consequently, avalanche frequency is assumed to be a constant parameter for a specific location, because each winter the probability of an avalanche with a certain frequency at that location is the same. But avalanche frequencies are vulnerable to the changing climate, as changes in climate patterns will result in changes to avalanche frequencies.

Avalanche frequency depends on position within an avalanche path, which is addressed differently for different ATES applications. For specific routes where the exposure is known (i.e., ATES $_{linear}$) the expected frequency of avalanches reaching the route is used, whereas for ATES zoning applications, frequency is used to define positions within the track and runout zones (i.e., higher frequency avalanches stop higher in the runout zone or track than lower frequency avalanches).

Table 7: Avalanche frequency terminology and associated frequency values and ranges used in ATES.

| Avalanche frequency | Average return period (years) | Average frequency (avalanches: years) | Annual probability of occurrence | Frequency range | Frequency descriptors |
|---|---|---|---|---|---|
| Very high | 0.3 | 3:1 | 1.0 | >10:1 to 1:1 | An avalanche occurs multiple times per year |
| High | 1 | 1:1 | 1.0 | 1:3 to 3:1 | An avalanche typically occurs once per year |
| Medium | 3 | 1:3 | 0.33 | 1:10 to 1:1 | An avalanche occurs every few years |
| Low | 10 | 1:10 | 0.10 | 1:30 to 1:3 | An avalanche occurs every 3 to 30 years |
| Very low | 30 | 1:30 | 0.03 | 1:100 to 1:10 | An avalanche occurs every 10 to 100 years |
| Extremely low | 100 | 1:100 | 0.01 | 1:300 to 1:30 | An avalanche rarely occurs |

This also has important implications for dry climates, where avalanche frequencies are typically lower than in wetter climates and thus the ATES ratings will be lower to reflect the lower long-term frequency in dry areas. ATES for Waterfall Ice Climbing

(Table 2) relies heavily on avalanche frequency assessments due to the problem of overhead hazard associated with this activity.

Table 8: The destructive avalanche size classification system (CAA 2024).

| Size | Destructive potential | Typical mass (t) | Typical path length (m) | Typical deposit volume (m³) | Typical impact pressure (kPa) |
|---|---|---|---|---|---|
| 1 | Relatively harmless to people | <10 t | 10 m | 50 | 1 |
| 2 | Could bury, injure or kill a person | $10^2$ t | 100 m | 500 | 10 |
| 3 | Could bury and destroy a car, damage a truck, destroy a wood frame house, or break a few mature trees | $10^3$ t | 1,000 m | 3,000 | 100 |
| 4 | Could destroy a railway car, large truck, several buildings, or a forest area of approximately 4 hectares | $10^4$ t | 2,000 m | 25,000 | 500 |
| 5 | Could destroy a village or a forest area of approximately 40 hectares | $10^5$ t | 3,000 m | 300,000 | 1000 |

The *magnitude* of a natural hazard is related to the energy released by the event. It is distinguished from *intensity*, which is related to the effects at a specific location or area (Jackson, 2013). Avalanche magnitude considers the destructive potential of the avalanche and is defined according to the Canadian avalanche size classification system (Table 8). Magnitude is inversely related to frequency because large destructive avalanches occur less frequently, while smaller ones occur on a more regular basis. Magnitude and frequency are also co-related to a specific location in an avalanche path. For example, a location near

the bottom of an avalanche path will be affected by larger avalanches less frequently, relative to a location higher in the same path.

### 4.2.6 Starting zone size and density

Increasing exposure to avalanche starting zones increases the severity of the terrain rating due to a higher likelihood of triggering or getting caught in an avalanche. In the ATES Technical Model, starting zone size is described in terms of the

potential size of avalanche release, whereas starting zone density refers to the number of starting zones within the area or along the route being assessed. This is particularly important with respect to route options and overhead hazard.

The number of starting zones, their size and proximity to the route all influence the overall ATES rating. Exposure to an isolated, single starting zone is usually less severe than exposure to multiple starting zones, but this would depend on their size and frequency. Overhead hazard (Figure 8) presents an additional challenge, particularly as the exposure becomes higher in

the avalanche path and closer to the starting zone. Remote or toe triggering of slopes is an important consideration when the exposure occurs below or to the side of the starting zone.

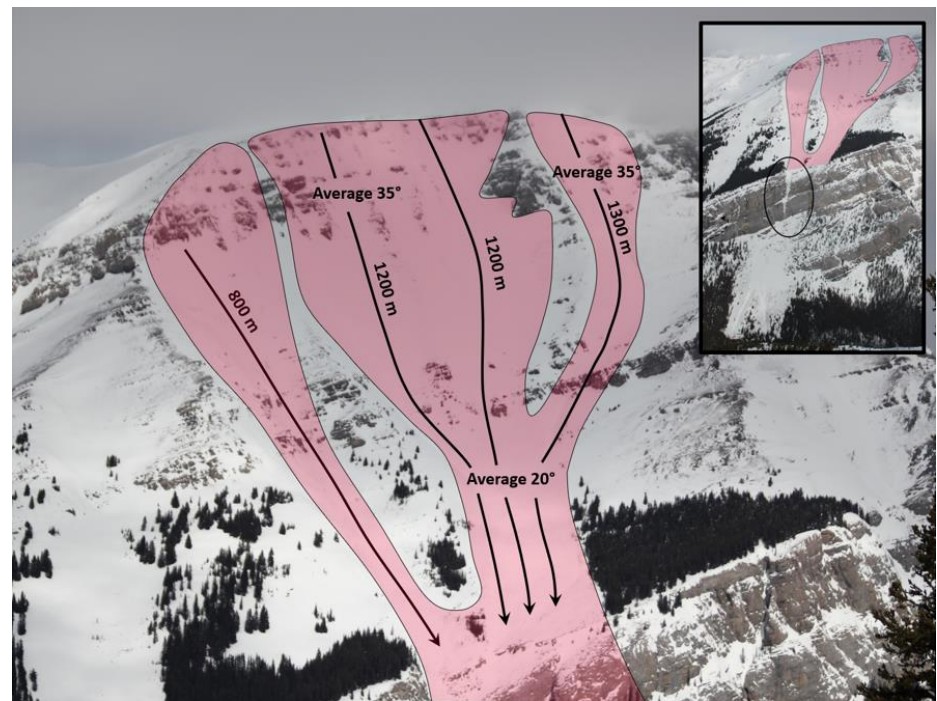

**Figure 8: Multiple large avalanche starting zones and tracks converge to create significant overhead hazard above the ice climb Bourgeau Left-Hand (inset) in the Canadian Rockies. This route is rated ATES Class 4 – Extreme terrain due to the overhead**
**hazard, > 1:1 frequency for > size 2 avalanches, and the possibility of human triggering while enroute.**

### 4.2.7 Runout zone characteristics

Runout zones are the lowest portion of an avalanche path, beginning below the track and extending downslope to the maximum extent of the avalanche path. This is where avalanches begin to decelerate, and deposition of snow and entrained material occurs. Certain terrain attributes effect the degree of avalanche exposure within runout zones. Characteristics such as runout
zone shape (e.g., abrupt transitions and confinement), terrain obstacles, and ground roughness influence avalanche runout behaviour, while proximity to starting zones, interconnectedness, and surface features influence the potential for remotely triggered avalanches. A remotely triggered avalanche occurs when a crack is initiated and propagates into adjacent terrain before causing a slab to release.

The ATES Technical Model (Table 3) considers two avalanche risk scenarios in runout zones: 1) being struck by a natural
avalanche starting overhead, and 2) remote triggering an avalanche by propagating a crack upslope into the starting zone where an avalanche releases. Every runout zone exposure scenario is unique, from simply crossing through the runout zone to travelling up the middle of it, directly under the avalanche track.

The ATES Technical Model describes exposure to runout zones on a continuum starting with Class 1 Terrain having smooth, well-defined runouts with no connection to starting zones above (Figure 9), ranging to Class 3 and 4 Terrain where runout
zones are overlapping, steep, confined, or contain terrain traps such as cliffs or crevasses. Class 3 and 4 runout zones may also have the potential for propagating remote avalanches into adjacent or overhead starting zones.

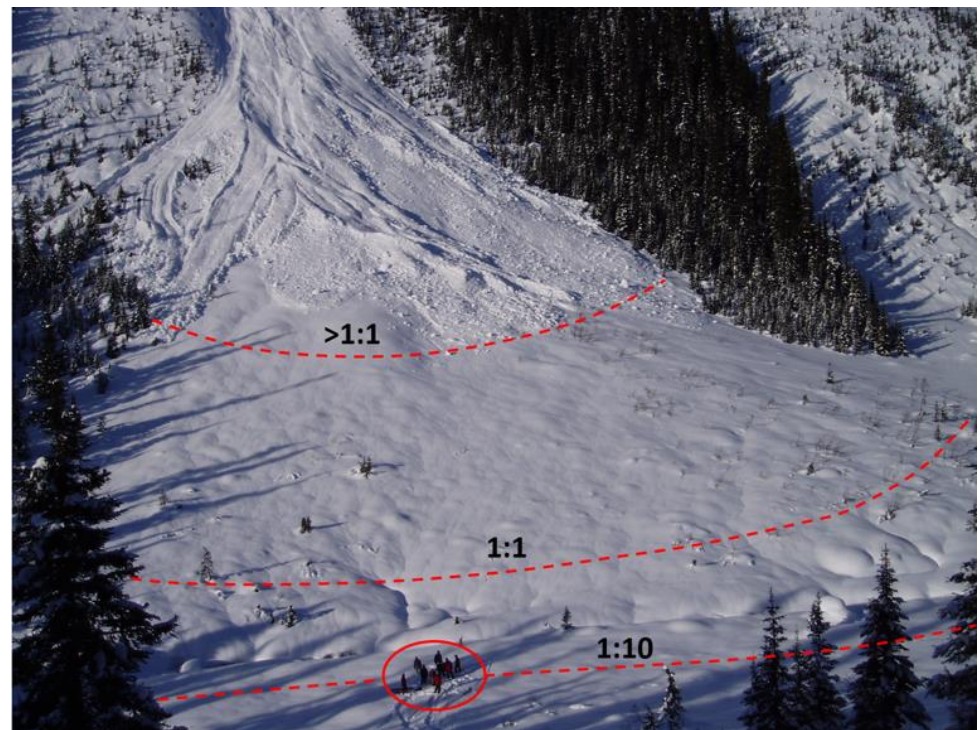

**Figure 9: An avalanche runout zone with a smooth surface, well-defined boundaries, and no potential to propagate into a nearby starting zones. Estimated avalanche frequencies are shown, indicating that the helicopter pickup location (red circle) is exposed ~1:10 years for > size 3, which makes this location ATES Class 2 – Challenging terrain.**

### 4.2.8 Route options

Route options are different ways to travel through the terrain and typically, every option presents a different level of exposure to avalanches, thus a different level of risk. Terrain with route options allows for different route-finding choices (Figure 10a), facilitating good risk management under various conditions. This contrasts with terrain that has limited or no route options, where people can be forced into terrain that will increase their risk (Figure 10b). Understanding and assessing route options is a crucial backcountry travel skill that occurs continuously from the planning stage right through to execution. Accordingly, route options is one of the most important input parameters to ATES, simply because optional exposure is much less committing than mandatory exposure.

Assessing route options depends on what is being assessed: a specific, predetermined Route or Corridor (ATES $_{linear}$), or an Area or Zone of terrain with no fixed route (ATES $_{spatial}$). Class 0 terrain avoids all avalanche terrain, Class 1 terrain can have many route options, some with no exposure, Class 2 terrain may be exposed to significant avalanche terrain, but options will exist to avoid it, Class 3 terrain has limited options with avoidance not possible, and Class 4 terrain forces mandatory, often extended exposure.

Basic risk management principles imply that when the avalanche hazard is High, backcountry users should choose routes with low avalanche terrain exposure to reduce risk; conversely, when the avalanche hazard is Low, choosing routes with a higher

avalanche terrain exposure may be an acceptable risk (Haegeli and McCammon, 2006). For some people though, higher levels of avalanche terrain exposure (or any avalanche terrain exposure) is never an acceptable risk, and in this case the presence or absence of route options is crucial information, especially the option to avoid avalanche terrain completely (i.e., Class 0).

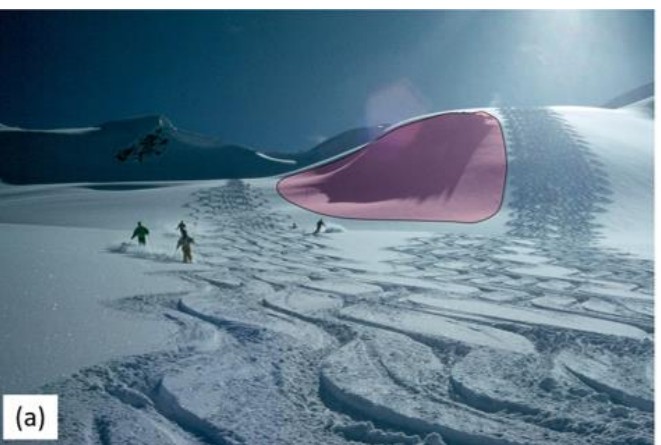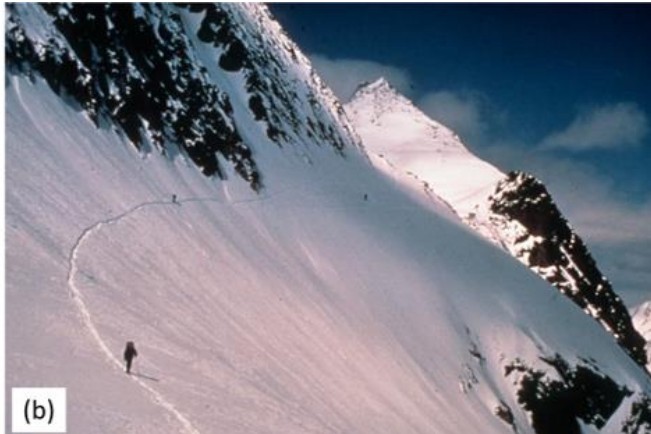

**Figure 10: Image (a) is ATES Class 2 because options exist to avoid avalanche paths, whereas image (b) is ATES class 3 because there are limited options to reduce exposure and avoidance is not possible; one must travel above a cliff to complete this route.**

### 4.3    Signal words, colours and numbers

To provide options for communicating ATES ratings to different audiences and to meet accessibility objectives, the system uses a combination of signal words, colours and numbers unique to each rating level (Table 9). Depending on the approach
(Table 10) and the channels of communication (e.g., digital, map or paper), different combinations of colours, words and numbers can be used to reach the target audience and to ensure inclusion and accessibility for all users of ATES.

Signal words are single terms that are used to denote the overall level of hazard implied by a warning (Hellier and Edworthy, 2006). They draw attention to a sign or label and quickly communicate the level of hazard. For ATES, each signal word is associated with a number which serves as multilingual label. While numbers are helpful in a multilingual environment, they
can be wrongly interpreted to hold some specific value or to imply linear growth between levels, which is incorrect. These numbers are simply labels.

Additionally, each rating level is assigned a unique colour for labels, lines or polygons on a map (Table 9). ATES colours were originally chosen to mimic the North American ski run difficulty system of green, blue, black (Statham et al., 2006) that is intuitive to North American users. European applications subsequently changed Complex terrain from black to red, to be
consistent with the ski run difficulty system in Europe. As a result, European ATES maps use different colours to represent Complex and Extreme terrain. ATES v.2 continues with the original colour scheme and adds white for Class 0 and red for Class 4 terrain (Table 9).

**Table 9: Signal words, numbers and colours associated with ATES.**

| ATES rating | Signal word | Colour | RGB code | Hex code |
|---|---|---|---|---|
| 0 | Non-avalanche | White | 255, 255, 255 | #ffffff |
| 1 | Simple | Green | 40, 201, 0 | #28c900 |
| 2 | Challenging | Blue | 0, 123, 255 | #007bff |
| 3 | Complex | Black | 0, 0, 0 | #000000 |
| 4 | Extreme | Red | 255, 1, 56 | #ff0138 |

However, warning system colours can present difficulties for people with colour vision deficiency (CVD) and not all colours work well when overlain on maps, especially when maintaining visibility of the underlying map reference layers is important. Polygon transparency settings must be chosen carefully to ensure the underlying basemap data remains visible. While black was originally a logical choice for Complex terrain because it is intuitive to skiers as a higher degree of terrain severity (Statham et al., 2006), this was before ATES became a mapping system. Today, black is a poor colour choice for displaying ratings on

some maps, as the basemap data is easily obscured and black lines can be difficult to distinguish on dark coloured mapping such as Google Earth (Figure 1).

Many warning systems in society use green and red, which provides a significant challenge for users with CVD. Engeset et al. (2022) tested six different colour combinations of ATES for conflicts with the avalanche danger scale colours and for users with CVD, recommending red for Complex, and black/red crosshatching for Extreme terrain. Huber et al. (2023) present an

ATES map for a test site in Austria using red for Complex and purple for Extreme terrain which shows the underlying basemap data well (Figure 11). Sykes et al. (2024) tested colours using a colour blindness simulator (Colblinder, 2024), and updated the ATES colour codes to improve accessibility (Table 9).

In order to communicate with a diverse audience, including those with CVD, ATES v.2 uses a combination signal words, numbers and colours to provide options for different ways to communicate with different receiver groups. Computers, websites

and digital products can use colourblind filters which help with deuteranopia, protanopia, and tritanopia. The design of an updated colour palette for ATES remains an open research question and user testing is necessary to determine a colour standard that achieves the best balance of comprehension, base map visibility and CVD compliance.

No single scheme works for all target audiences. Applying a suitable combination of colours, numbers and signal words in combination with an accessible legend is likely to achieve the best results.

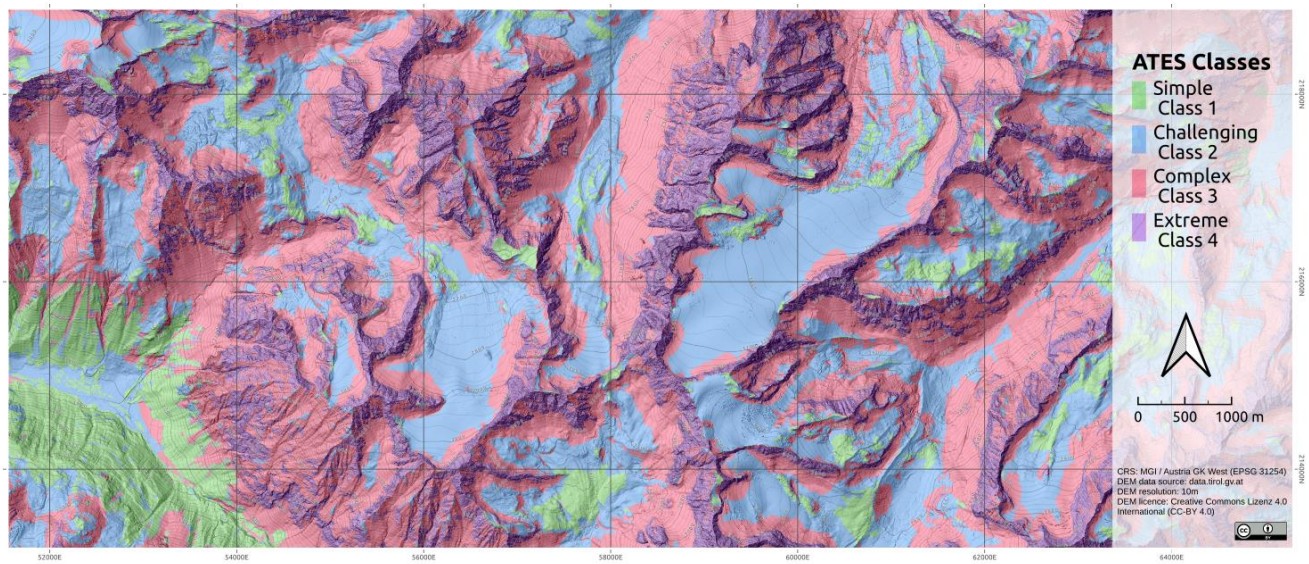


**Figure 11: An AutoATES map produced for a test site in Austria (Huber et al., 2023) that explores different colour patterns for Complex (red) and Extreme (purple) terrain.**

### 4.4    Target audience

A thorough understanding of the receiver, the target audience, is necessary for effective risk communication. Laughery and
Brelsford (1991) implored warning designers to "know thy user" with regard to (1) demographics and age, (2) familiarity with
the product, (3) competence (technical knowledge, language, reading ability) and (4) hazard perception.

The ATES system has three distinct target audiences:

1. Avalanche professionals, educators, mappers, and guidebook authors
2. Backcountry recreational travellers: skiers, snowboarders, snowmobilers, snowshoers, climbers and hikers
3. Backcountry workers: persons employed to perform work in avalanche terrain

The Technical Model (Table 3) is designed for avalanche professionals, mappers or guidebook authors to use its specifications
to assess avalanche terrain, determine the exposure of people to that terrain, and produce an ATES rating. The Technical Model
also targets avalanche educators, who can use the model's specifications for teaching the specific elements of avalanche terrain,
how each is scaled, and how they interact with the exposure of people to determine the severity of avalanche terrain exposure.
The Communication Model for backcountry travel (Table 1) is targeted at all backcountry users who move through avalanche
terrain, regardless of recreation type. The language gives simple advice on expectations of exposure and potential options for
mitigating risk. ATES is analogous to the avalanche danger scale (Statham et al., 2010; Avalanche Canada, 2022; EAWS,
2024) and targets the same audience, including workers (often industrial/resource staff) who follow rules-based workplace
safety practices.

The Communication Model for waterfall ice climbs (Table 2) targets winter ice climbers and focuses on the concepts of exposure time, avalanche frequency, human-triggering in terrain traps and options to reduce exposure. The system has recently been applied to Avalanche Canada's ice climbing avalanche atlas (Statham and Heuniken, 2023).

## 5 Application of ATES

The application of ATES starts by considering the objectives of the final product, which informs the approach to assessment

and communication methods. The objective and approach depend on the target audience, their intended use of the terrain ratings and the availability of terrain data.

For example, the objective might be to facilitate recreational trip planning, in which case a single ATES rating for a specific area or route might be sufficient, or multiple rating segments along that route for a more precise assessment. However, a navigational aide for backcountry travellers would typically require high-resolution ATES zones or specific route segments.

Over the past two decades of ATES use, four distinct approaches to ATES classification have emerged (Table 10). An *Area* defines the boundaries of an overall assessment and can be given either a single rating (Figure 12a, b) or broken down into smaller scale zones (Figure 12c). A *Route* defines a linear path of travel from start to finish (Figure 12d) and can be broken down into shorter route segments using lines to represent precise routes, and polygons to represent a *Corridor* of travel where navigational freedom is possible (Figure 12b). A *Zone* is a specific slope or grouping of terrain features with common ATES

characteristics that uses a polygon to spatially represent the zone, typically surrounded by adjacent polygons showing their ATES zone ratings (Figure 12c).

Table 10: ATES approaches showing feature types and their spatial representation (Sharp et al., 2023).

| ATES Feature | Example Application | Spatial Representation |
|---|---|---|
| Areas | Rating commonly defined region with either a well-defined geographic boundary or an ambiguous one | Point (Figure 12a) or polygon |
| Zones | Rating a specific slope or terrain feature within a well-defined geographic boundary where ATES parameters dictate the zone boundaries | Polygon (Figure 12d) or raster (Figure 13) |
| Corridors | Rating a physical or conceptual path of travel between defined starting and end points with navigational freedom within a well-defined geographic boundary or an ambiguous one | Polygon (Figure 12c) or line |
| Routes | Rating a physical or conceptual path of travel between a defined starting and end point with limited navigational freedom | Line (Figure 12b) |

The major difference between these approaches is that ATES ratings for routes rates the *actual* terrain exposure of specific,
pre-determined Routes or Corridors, such as an ice climb or ski tour where the start, route and endpoint are known. ATES
ratings for Areas or Zones rates the *potential* terrain exposure because a specific route is not prescribed, such as an open alpine
bowl with numerous different ski lines. In this case, once a route has been planned through the terrain, then the actual exposure
can be evaluated and related to the ATES ratings.

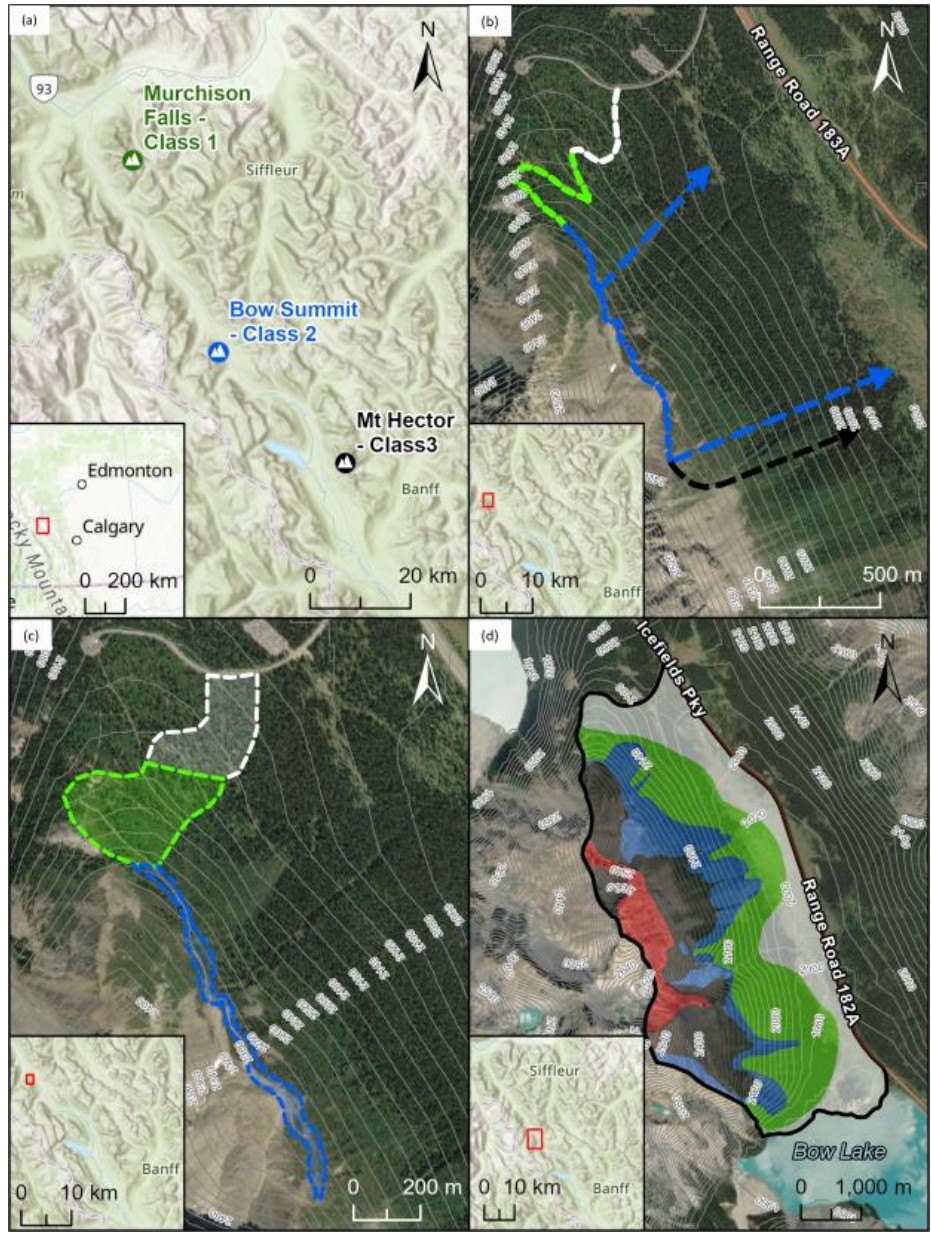

**Figure 12: Spatial representations of different ATES feature types illustrating Areas represented as single-rating points (a); multi-
rating routes represented as lines (b); multi-rating Corridors represented using polygons; and ATES zones represented using
polygons. Basemap source: Natural Resources Canada, Esri.**

The major difference between these approaches is that ATES ratings for routes rates the *actual* terrain exposure of specific, pre-determined Routes or Corridors, such as an ice climb or ski tour where the start, route and endpoint are known. ATES
ratings for Areas or Zones rates the potential terrain exposure because a specific route is not prescribed, such as an open alpine bowl with numerous different ski lines. In this case, once a route has been planned through the terrain, then the actual exposure can be evaluated and related to the ATES ratings.

## 5.1 Spatial scale

Spatial scale refers to the size or extent of a geographic area. Table 11 describes spatial scales used in avalanche forecasting
(Statham et al., 2018), and these scales also relate directly to avalanche terrain assessments.
It is important to determine at what scale the ATES ratings are being applied at, and recognize that not all ATES criteria shown in Table 3 can be applied at all scales. Parameters such as starting zone size and density, runout zone characteristics and exposure require multiple slopes in order to assess, meaning they often cannot be applied to single slopes or terrain features. Forest density, in contrast, works better at smaller scales where there is less variation across the terrain and the density can be
determined more reliably. In some scenarios, small scale (e.g., terrain feature) zoning will not be required, in which case a larger scale can be applied. To achieve a larger scale, ATES mappers must filter out terrain features or route segments that are below the target scale, and group these features together into larger scale zones or routes.
For example, when classifying a pre-determined route, the scale of the entire Area is already defined by the route. However, along that route there will be variations in avalanche exposure. These could be represented using smaller scale ATES ratings
for improved accuracy, or they could be grouped together as part of the whole route and a single rating issued. Single ratings for Routes and Corridors should default to the highest terrain class along the route. Similarly, while an overall rating of Class 3 could be assigned to an Area, within that Area there could be Zones of Class 1 and 2 terrain. Single ATES ratings for Areas and Zones sometimes default to the highest rating level, but this depends on the scale of the ratings, and whether there are route options within that Area. For example, while the overall Area may have some Class 3 terrain, if there are options to avoid
it, then the rating is Class 2.
The smaller the scale, the higher the resolution and more precise the classifications will be, but this comes at the cost of greater effort and resources. To be accurate enough to be used as a real-time navigational aid, a spatial scale of at least 20-30 m (i.e., terrain feature) is required (Larsen et al., 2020).


**Table 11: Spatial scales for ATES assessments (Statham et al., 2018).**

| Spatial Extent | Description | Examples | Scale |
|---|---|---|---|
| Terrain Feature | Individual geographic features contained within a larger slope | Convex roll, gully or terrain trap | Micro $< 1 \text{ km}^2$ |
| Slope | Large, open, inclined areas with homogenous characteristics bounded by natural features such as ridges, gullies or trees | Typical avalanche starting zone or wide-open area on a ski run | |
| Path or Run | Multiple interconnected slopes and terrain features running from near ridge crest to valley bottom | Full length avalanche paths with a start zone, track and runout zone or typical long backcountry ski run | |
| Mountain | An area rising considerably above the surrounding country with numerous aspects and vertical relief running from summit to valley bottom | Ski resort area or typical single operating zone in a snow cat skiing area | Meso $> 10^2 \text{ km}^2$ |
| Drainage | An area with a perimeter defined by the divide of a watershed | Typical single operating zone in a helicopter skiing area | |
| Region | A large area of multiple watersheds defined by mapped boundaries | Typical public forecasting area or public land jurisdiction | Synoptic $> 10^4 \text{ km}^2$ |
| Range | A geographic area containing a chain of geologically related mountains | Mountain ranges or sub-ranges | |


## 5.2    Assessment methods

Evaluating avalanche terrain exposure using ATES requires qualified people skilled in avalanche terrain assessment and backcountry route-finding. Assessors with local terrain and route familiarity is a significant asset and necessary to analyse the interaction between people and avalanche terrain. Local knowledge of trails, backcountry routes or climbs is an essential input
in lieu of pre-mapped routes.

Rating avalanche terrain using ATES can be straightforward for single routes with single ratings. For uncomplicated terrain with good data, such as one well-travelled trail with only a few avalanche paths or an alpine bowl with high quality mapping and imagery, sufficient accuracy can be achieved without field surveys or complex analyses. For more complicated projects such as large areas with extensive avalanche terrain, unfamiliar travel routes, significant overhead hazard or a need for small
scale ATES zones, a more rigorous approach and level-of-effort is necessary. Typically, this utilizes some combination of GIS analysis, field investigations, aerial photographs, satellite image interpretation, as well as climate analysis and runout estimation.

Data for the analysis is collected using various methods, both qualitative and quantitative. GIS analysis provides a deterministic evaluation of some ATES parameters and helps to reduce human bias (e.g., Delparte, 2008; Campbell and Gould, 2013; Toft

et al., 2024), but not all ATES parameters can be represented digitally. Realistically, only slope angle and forest density can be determined objectively, given adequate resolution, leaving the remaining ATES parameters to be mostly a subjective assessment. Route options and exposure both require a location on the ground to assess, and this means evaluating either a predetermined route or a conceptual line through the terrain. Data for every Technical Model parameter shown in Table 3 is not often available, so the assessor must make do with the best information they can obtain. ATES is intended to be used by

both field practitioners as well as desktop GIS specialists, and ideally a team of both. Assessors ultimately develop their own techniques and work within the bounds of their organization's capacity, but the most accurate results are achieved through a collaborative approach.

Ratings are determined by analysing the terrain against each ATES parameter for best fit, comparing to the levels above and below, then determining what the best overall ATES rating is. The following five-step process guides the determination of an

ATES rating:

For every Area, Route, Corridor or Zone:

1. Assess each Technical Model (Table 3) parameter independently and determine its rating level
2. Determine which (if any) default criteria are met (this determines the minimum rating level)
3. Compare each of the remaining terrain criteria to the minimum rating level or higher

4. For criteria higher than the minimum rating level, determine if this outweighs the minimum rating level to determine the ATES rating
5. Compare this to the Communication Models (Tables 1 and 2) for coherence

For manual assessments at micro and meso scales (Table 11), ATES ratings and mapping should be reviewed and field checked by peers familiar with the terrain. For zoning avalanche terrain exposure at synoptic scales, such as regional or mountain range

mapping, manual assessments and field checking for verification of the entire Area is often not practical and instead, a targeted approach to the field work, or an automated classification approach (or a combination of both) is often necessary.

### 5.3 AutoATES

Automated avalanche terrain classification enables large areas of mountain terrain to be analysed and coded by a computer algorithm (Figure 13). This significantly reduces the cost of producing ATES ratings, improves consistency, and makes the

system more accessible. Larsen et al. (2020) developed AutoATES v1.0, which was used to produce ATES zone maps for all of Norway using only a digital elevation model (DEM) as input. AutoATES v.2.0 (Toft et al., 2024) has been updated to match the ATES v.2 model presented in this paper, and the algorithm's performance has been improved to better handle forest data, overhead exposure and flat runout zones.

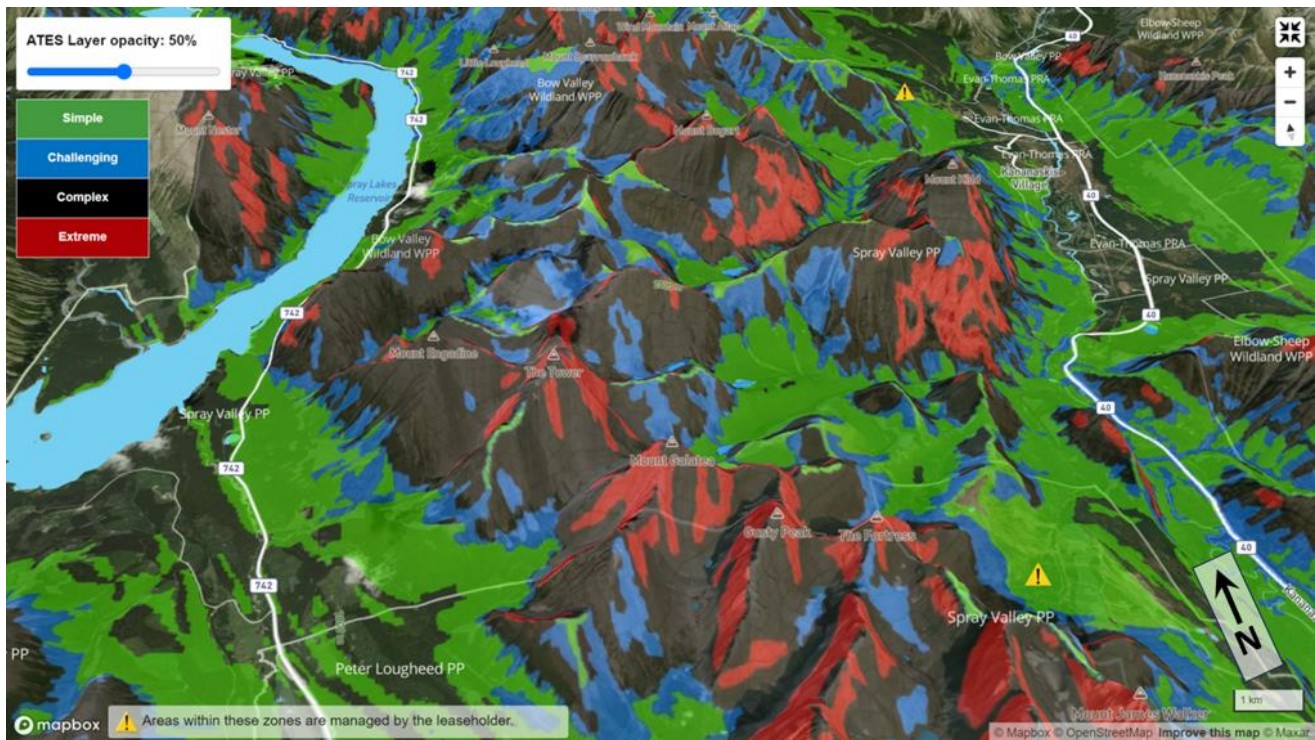

**Figure 13: AutoATES mapping of ~450 km² in Kananaskis Country, Canada (Alberta Parks, 2024). In this example, the ATES layer opacity can be adjusted to improve the visibility of the base map data, and Class 1 terrain includes Class 0.**

AutoATES mapping can be adapted to local conditions by tuning the model parameters based on feedback from avalanche experts. Sykes et al. (2024) performed validation testing on AutoATES v.2.0 in Connaught Creek and Bow Summit areas of Canada. Manual ATES zone "benchmark maps" for each area were developed collaboratively by three field experts. The

benchmark maps were used as a validation dataset to tune the input parameters of AutoATES to the local characteristics of each study area. AutoATES v.2.0 maps were then produced for the same areas, compared to these benchmark maps (Figure 14) and found to agree with 74.5% of Connaught Creek and 84.4% of Bow Summit ATES ratings (Sykes et al., 2024).

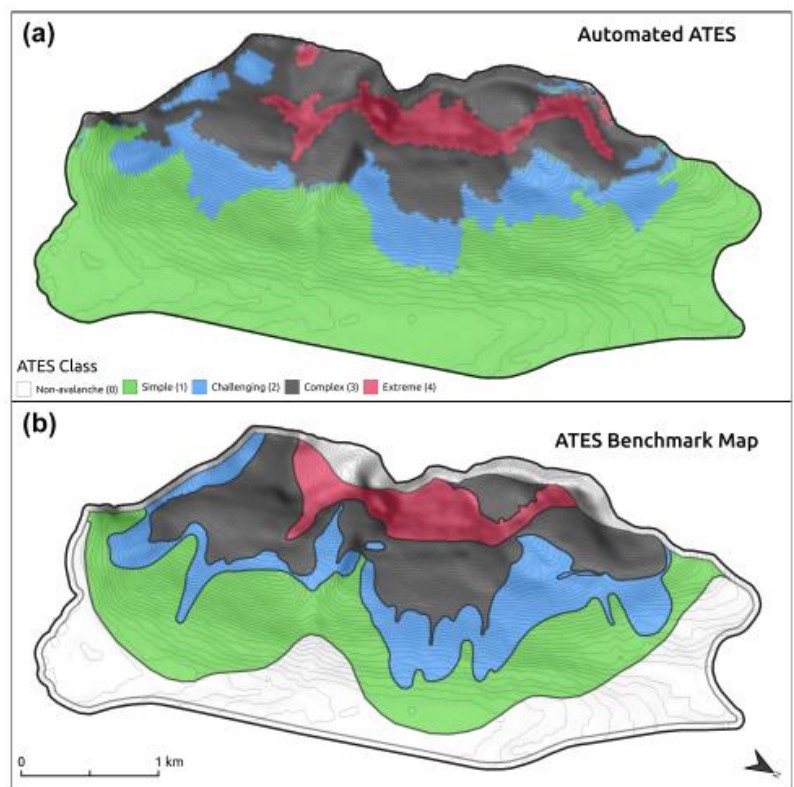

**Figure 14: A validation study comparing manual versus automated ATES mapping (Sykes et al., 2024) where the AutoATES map agreed with 84.4% of the manual "benchmark" map.**

One of the biggest advantages of automated ATES zone mapping is that it can downscale zones to a much higher resolution than is practical with manual mapping. While it is possible to manually downscale to smaller zones, this requires a level-of-effort that may not be cost effective, particularly in synoptic or meso scale areas (Table 11). This limits the scope of manual mapping in comparison to automated mapping, which can cover entire mountain ranges consistently, and at smaller scales.

## 5.4 Presentation

ATES ratings can be displayed visually on maps or marked-up photos as Areas, Zones, Corridors or Routes (Table 10; Figure 12). Coloured lines and/or transparent polygons with fuzzy set boundaries can illustrate ATES ratings, ideally with the underlying ATES terrain attributes stored (Sharp et al., 2023). Fuzzy set boundaries incorporate uncertainty by overlapping and fading the boundary between adjacent ATES polygons, indicating that the boundary is not a precise line but rather an area of transition.

In addition to maps, ATES ratings for specific routes can be communicated using words, numbers and colours. Backcountry recreation guidebooks, brochures and online information commonly use textual ATES ratings as an adjunct to a detailed route description, map and other important information about a specific route.

## 6    Limitations

ATES is an avalanche terrain assessment and communication system that relies heavily on expert knowledge and judgement (Toft et al., 2024). Despite developments to make it more deterministic (Campbell and Gould, 2013), applying and using ATES remains primarily an exercise in judgement that requires ground truthing and peer review. Although ATES incorporates the terrain parameters necessary for avalanche experts to capture their interpretation of the avalanche terrain, interpretations vary between individuals and can lead to inconsistency in application, i.e., two experts rating the same avalanche terrain using

ATES may have different results. These differences highlight the subjectivity in manual ATES ratings and the challenge of having multiple individuals produce consistent ATES ratings (Sykes et al., 2024, Schmudlach and Köhler, 2016b). Manual interpretation of geospatial data combined with observed terrain parameters is a time-consuming process which limits the scope of manual ATES mapping to high-traffic areas such as popular recreation areas and pre-defined worksites. ATES ratings for a specific route is less time consuming since the assessment focusses on a predefined line or corridor where the

exposure is known, rather than all terrain in an area where the exposure varies. In these areas, costs can be justified relative to the large number of backcountry users (Larsen et al., 2020; Sykes et al., 2024) and terrain familiarity of local experts, but this is impractical for large swaths of mountainous terrain. Synoptic scale ATES zone mapping is not practical using manual methods, and the development of AutoATES (Toft et al., 2024) has been an important step towards enabling a broader implementation.

While ATES zone maps illustrate potential exposure across landscapes, the receiver of the information cannot assess their actual exposure until a location, or route is specified. Once the receiver plans a route on the map (explicit or conceptual), or uses blue dot navigation, then a location becomes evident, and the ATES ratings can be related to that spot. Modern digital mapping applications that enable route planning are well suited to include an ATES layer, whereby the user can draw their route on the map and then turn on/off an ATES layer to see how that route intersects with the ATES ratings.

It is important to be aware of the limitations and uncertainties associated with using digital elevation models (DEM) to produce avalanche terrain maps. Research confirms that starting zone and runout zone modelling is sensitive to DEM type and resolution (Bühler et al., 2011), and that high resolution DEM (i.e., <5m) is ideal for capturing terrain features relevant for avalanche release (Bühler et al., 2018). But high resolution DEM have limited availability worldwide, and 5m DEM is not always necessary for modelling avalanche terrain exposure. Currently, 10m satellite imagery and 30m DEM data is available

worldwide, for no cost. Sykes et al. (2024) found that the resolution and type of input DEM does not have a large impact on the overall accuracy of the AutoATES model.

Finally, developers of publicly available, digital avalanche risk applications must be wary of the potential for dangerous errors when their applications combine micro scale, high resolution DEM with synoptic scale, low resolution avalanche bulletin information. Generalized aspect/elevation diagrams broadly applied at synoptic scales by avalanche forecasters is a mismatch

with high resolution DEM terrain models, and this type of scale mismatch will produce errors which are easily masked by the ease of use and perception of accuracy on a mobile phone application.

## 7    Conclusion

Terrain rating systems play an essential risk management function in recreational outdoor activities such as climbing, hiking, kayaking, skiing, and biking. Industries where workers are exposed to avalanche risk also rely on terrain rating systems to enable occupational health & safety policies. Combined, these systems have helped millions of users plan and execute their activities by simplifying complex terrain attributes into easily understood categories that can be used to manage risk and improve the experience.

Backcountry avalanche risk is a complex interaction between snowpack, terrain and people, where terrain is the only factor that is constant over time. It is often said that "when snow is the problem, terrain is the solution" and for decades professional mountain and ski guides have considered terrain assessment and route selection to be the principal mitigating factor in backcountry avalanche risk management: when nothing is exposed, nothing is at risk.

But communicating to a lay person on how to evaluate avalanche terrain and manage their risk in the backcountry is challenging, as the subject is complex with many technical variables that are easily lost upon the target audience. The classic slope-angle based terrain choice method (Landrø et al., 2020) which has dominated avalanche decision making strategies for decades, is limited in its scope. Its attraction is that it's easy to understand, measurable and accurate for human triggering inside avalanche starting zones. But slope angle alone does not account for important factors such as propagation, overhead hazard, terrain traps, avalanche frequency and the concept of exposure in general, which are the fundamentals of backcountry avalanche risk management. Encompassing them into a simple, five-level terrain classification system that can be easily understood by the receiver is important in the same way that the avalanche danger scale (Avalanche Canada, 2022; EAWS, 2024) helps people to categorize and understand the level of avalanche danger in a simple way.

Public avalanche bulletins warn about backcountry avalanche danger, which is constantly changing and carries uncertainty, but this is only part of the avalanche risk equation. Ultimately, people choose their own risk by making decisions about where, when, and how they travel. Even during periods of High avalanche danger, a simple reduction in exposure can reduce or eliminate the risk. On small-scale terrain features, even minor adjustments in how one is exposed to the danger will change their risk – a few meters in either direction can be the difference between a low and high-risk situation. Thus, controlling terrain exposure is the most important avalanche risk management skill necessary for winter backcountry travel, and the objective of ATES is to make that more explicit and easier to understand for backcountry users.

ATES began in 2004 as a simple avalanche terrain rating system for specific backcountry ski tours, intended for trip planning and implemented in response to an avalanche disaster in Canada's Glacier National Park. Soon after, ATES was used to rate avalanche exposure on waterfall ice climbs, and by 2010 ATES ratings were being mapped into zones using basic GIS. In 2020, the AutoATES algorithm enabled landscape scale mapping of ATES ratings, enabling more accessible, widespread ATES mapping. Today, AutoATES technology can be used to automatically classify large, synoptic scale areas, while manual ATES methods can be applied to smaller scale projects or for specific routes, where the input and accuracy of the human touch is necessary.

This paper introduces ATES v.2, which builds on 20 years of operational experience using ATES as a risk management tool in avalanche safety practices for public recreation and workplace avalanche safety. The updated five-level ATES adds Class 0 – Non-Avalanche terrain and Class 4 – Extreme terrain to the original three-level system. Additionally, ATES v.1/04 and the ATES Zoning Model have been combined into a single Technical Model for assessment, with two corresponding Communication Models for backcountry travel and waterfall ice climbing. Using ATES v.2, avalanche terrain exposure can

be mapped as Areas, Zones, Corridors or Routes (Table 10). Alternatively, specific routes can be given a terrain rating, or series of ratings, to accompany a route description in the same way that rating systems are used for rock climbing and whitewater.

## 8 Code availability

AutoATES is open source, and the model code is available via Zenodo (https://doi.org/10.5281/zenodo.10712035, Toft et al.,

2024). The data to replicate the AutoATES validation methods in Sykes et al. (2024) are available in an Open Science Framework repository (https://doi.org/10.17605/OSF.IO/ZXJW5, Sykes et al., 2024).

## 9 Author contributions

GS was the original creator and author of ATES v.1/04 and led the implementation of Parks Canada's initial application of ATES to backcountry tours and waterfall ice climbs in Canada's national parks. CC developed the ATES Zoning Model and

was the first to apply ATES ratings on maps through his work with Avalanche Canada. Both authors have continued to develop and apply the ATES method to recreational and workplace applications, and both helped to develop the validation dataset for AutoATES. GS led the development of this manuscript with support from CC.

## 10 Competing interests

Both authors declare that they have no conflicts of interest.

## 11 Acknowledgements

The authors wish to thank Edward Bair, Stephan Harvey, H.P. Marshall and one anonymous reviewer for their helpful peer review. As well Chris Argue, Rune Engeset, James Floyer, Brian Gould, Pascal Haegeli, Steve Holeczi, Sarah Hueniken, Alan Jones, Karl Klassen, Mike Koppang, Eirik Sharp, Andy Sovick, John Sykes and Håvard Toft for their contributions to this work.

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
