# Peer review of "The Avalanche Terrain Exposure Scale (ATES) v.2"

_Natural Hazards and Earth System Sciences, 2024_

## Author Comment (AC1)

We would like to thank Edward Bair for his efforts to read our manuscript and provide constructive feedback that will undoubtedly improve this paper. We have reviewed the feedback carefully and offer our responses below.

Our comments are listed in red text and are numbered 1-16.

**ATES Paper Peer Review – Edward Bair – 06 July 2024**

The Avalanche Terrain Exposure Scale (ATES) v2 is an update to a 20-year-old system introduced in Canada as a risk management tool for backcountry travel. This new version of ATES offers improvements that combine developments over the past two decades into a single model.

After reading the manuscript along with the original ATES model article (Statham et al., 2006) and subsequent developments (e.g., Campbell and Gould, 2013), I've gained an appreciation for the deep refinement of ATES. However, I'm unclear about why, with version 2.1, there is a need for a peer-reviewed publication, when there was not for the original ATES nor its interim conference papers. Likewise, I have concerns about the re-use of previously published material. As outlined in the review criteria, publication in NHESS requires new work that is scientifically significant, like the original ATES. ATES v2.0 has some new features, but appears as an incremental improvements on two decades of work. A figure or table summarizing the evolution of ATES since its inception would be helpful. The requirement for a substantial new advance is why there are often not peer-reviewed manuscripts for new versions of a model. There are examples (e.g., Sturm et al., 1995; Sturm and Liston, 2021; Dozier, 2022; Dozier et al., 1981), but the authors must convince the reviewers that the changes are substantial enough to change the understanding of the topic. Conversely, given that the previous ATES work has been published mostly in conference proceedings, there are issues with preservation. For example, the "permalinks" for ISSW Proceedings are not an authoritative archival method, whereas the Digital Object Identifier that is produced for this discussion article is. The authors should consider stating their motivation for a peer-reviewed article directly in the manuscript.

1. The reviewer is correct that ATES v1/04 and the ATES Zoning Model have both been published in ISSW conference proceedings in 2006 and 2013 respectively. As described in this manuscript, the ATES system has been widely adopted over the past 20 years (e.g.: Canada, United States, New Zealand, Norway, Sweden, Switzerland, Austria). Thus, a significant gap now exists in the literature when a risk management system that has been so widely adopted, has no peer-reviewed article as its baseline reference. Our objective is to fill this gap by publishing a modern, peer-reviewed baseline reference for ATES which corresponds with our revision to ATES v.2.

We are aligned with the objectives of this NHESS Special Issue, as our goal is to expand on our ATES v.2 ISSW paper (Statham and Campbell, 2023). By design, this means republishing and expanding on material that is in the ISSW conference proceedings. We quote from the NHESS website *"We invite all ISSW presenters interested in a peer-reviewed scientific publication to*

*submit manuscripts that explain their research in more detail than in the conference proceedings. To be considered, these papers need to be scientifically more rigorous, which means that they contain expanded technical information, additional details on methods, and a more complete description of results and conclusions."*

ATES v.2 is a significant update to the model. It merges the previous two ATES models (Statham et al., 2006; Campbell and Gould, 2013) into a single ATES v.2, resolving conflicts that existed between them. It presents two important new terrain classification levels (Class 0 and 4), it introduces ATES for waterfall ice climbing (Table 2) which is new, and it removes the glaciation criteria, which was a defect in the original model.

The manuscript itself provides new, detailed explanations on each of the eight Technical Model (Table 3) criteria and defines commonly used terminology (Table 4, slope angle terminology) and field-based assessment techniques (Table 7, avalanche frequency) and introduces them into the literature. This will provide a baseline for future reference and developments. Figures 1, 2, 3, 4, 5, 6, 7, 8 and 10 are all original, previously unpublished figures.

Section 5, Application of ATES is all new information that has been expanded from Statham and Campbell (2023). None of the previously published conference papers (2006, 2013) contain any description of how to apply the ATES method. This manuscript introduces ATES feature types (Table 10) and gives graphic examples of mapping methods (Figure 10).

We submit that Figures 9, 11 and 12 are used (with permission) from other publications, but because this article will serve a baseline reference for the ATES method, we would be remiss not to describe important developments by other authors that are available today (Figure 9, colour choice and Figures 11 and 12, AutoATES).

We agree with the reviewer that the "permalinks" for ISSW Proceedings are not an authoritative archival method, whereas the Digital Object Identifier that is produced for this discussion article is. Our objective is to correct this gap for the ATES system and create a more robust reference.

The article contains reprinted material, sometimes verbatim. As cited, Tables 4-6 are from Campbell and Gould (2013) and Figure 12 appears verbatim in Sykes et al. (2024, Figure 10) as well as in parts of Toft et al. (2024, Figure 4). In fact, most of the figures and tables (Figure 9, Figure 11, Table 8, Table 10, Table 11) are from previous publications. At the least, these reproductions beg the question: what's new here?

2. Table 4 links commonly used slope angle terminology with slope angle values and is not from Campbell and Gould (2013).  The use of slope angle terminology such as "low angle" (Table 4) is ubiquitous across the avalanche forecasting and backcountry travel sector. ATES v.2 defines the meaning of this language by linking the terminology with slope angle values.

Tables 5 and 6 are adapted from Campbell and Gould (2013) and form an essential baseline for ATES mapping parameters, they cannot be left out. Table 5 was previously published as a footnote, and Table 6 has been adapted to include Class 4 terrain.

Figure 12 is verbatim from Sykes et al. (2024) but note that Statham is a co-author on this publication and both Statham and Campbell undertook the validation study that this paper (and Figure 12) describes. There have been three ATES publications through NHESS in 2024: Sykes et al. (2024), Toft et al. (2024) and this manuscript. While we have tried to eliminate overlap, some overlap is inevitable and necessary to holistically describe the work that has been done.

Figure 9 is used with permission and illustrates a unique colour choice and mapping approach done in Austria. Colour choice for ATES remains an area for future research, and this figure is a good example that was published in the ISSW 2023 proceedings. We will improve the captioning on this figure to explain the need for future research into ATES colours.

Figure 11 has not been previously published. This is an image of new autoATES mapping done by Alberta Parks and is on their website. This is the first public online publication of an AutoATES map in Canada and is published on the Alberta Parks website.

Table 8 is taken directly from CAA (2016) and is the baseline reference for avalanche size in Canada. This table has been unchanged for decades and is an essential reference point when describing avalanche magnitude.

Table 10 is all new and has only been published in Statham and Campbell (2023), which is the ISSW paper that this manuscript is expanding on for the special edition.

Table 11 is previously published in Statham et al., 2018 where it was developed for application to avalanche hazard assessments. The same scale breakdown has direct application to terrain assessments and thus the table is republished here as part of the Application section, which is new information.

Given that ATES is likely to be applied to remote areas, the AutoATES v2 open-source code is a significant new contribution. It is discussed in Section 5.2.1 and should have been included in a "Code availability" section (https://www.natural-hazards-and-earth-system-sciences.net/submission.html#manuscriptcomposition). Yet there is already a publication for AutoATES v2.0 (Toft et al., 2024). Thus, this section too contains previously published information.

3. We agree that the AutoATES v.2 open-source code is a significant new contribution, and we will improve our manuscript to a better direct the reader to the "code availability" found in Toft et al. (2024). However, our manuscript is not about AutoATES, although it is referenced, thus we have not developed any code to make available.

Not present, although maybe present in previous ATES literature which could have been cited, is a discussion of the limitations of the static hazard model. If there is no snow, there is no snow avalanche hazard, regardless of terrain. Given that the majority of backcountry use (i.e., hiking) comes in the summertime, that is a significant limitation. Further, as someone who has spent considerable effort mapping snow from satellites, I find the omission of the changing snowcover to be a major model shortcoming. A static map is certainly easier to produce, but the authors

could at least discuss how a dynamically modeled or remotely-sensed snowcover could be incorporated into the ATES model.

4. We agree that this manuscript should improve its description of static versus dynamic hazard models. We will expand on this following Line 29 where the concept of the static model is introduced, and we will describe two key references regarding static and dynamic hazard maps (Harvey et al., 2018, Schmudlach and Köhler, 2016).

We will also use the Introduction to better describe the avalanche risk framework that ATES fits within (Statham, 2008; Statham et al., 2018), which will include descriptions of the dynamic nature of avalanche hazard and its interaction with exposure and vulnerability to create risk. Reviewer #3 has suggested this, and we believe we can address this reviewer's concerns here too.

Static terrain maps are a foundation of dynamic hazard maps; an underlying DEM or even ATES map is necessary to combine with a dynamically modelled or remotely sensed snowpack to develop dynamic avalanche hazard maps. The whole point of ATES is to provide a static terrain exposure model with ordinal categories that can be easily interpreted for public recreation and workplace avalanche safety (Tables 1 and 2). ATES can integrate with dynamic avalanche hazard/danger models to give a kind of risk rating to assist with basic decision making. The Avaluator (Haegeli et al., 2006) was the first decision framework to integrate the changing snowcover with ATES ratings (rather than simply slope angle).

Schmudlach and Köhler (2016) integrated Swiss avalanche bulletins with static terrain maps to create dynamic avalanche hazard maps on their popular website https://www.skitourenguru.ch, first for Switzerland and now including most of central Europe. The Swiss terrain hazard maps developed by Harvey et al., (2018) are another example of static terrain maps that provides crucial input to dynamic avalanche hazard models.

We agree with the reviewer that dynamic avalanche hazard maps are important, and likely the future as snowpack models continue to develop, but they are based upon using static terrain data, and that is the role of ATES.

In short, I commend the authors efforts with ATES v2, however this manuscript requires considerable revision to justify how it is a new contribution. Currently, it appears to be a summary of previously published work rather than an original research article. I'm stopping short of rejecting the manuscript and suggesting it be resubmitted as a review article, but the authors should consider that route.

NB 6/5/24

Minor:

Abstract

What's new here? Consider adding l 530-536 and maybe l 37 – 40 to the Abstract

5. We will consider adding this to the abstract.

Introduction: Element-at-risk sounds like agency jargon. Define.

6. "Element-at-risk" is not agency jargon, but a term defined in Jamieson (2018) and widely used in natural hazards risk assessment (e.g.: Papathoma-Köhle et al., 2007; Sterlacchini et al., 2014). Assets-at-risk is a synonym. We will modify Line 21 to read "…where the elements-at-risk such as skiers, climbers, snowmobilers or workers and exposed to avalanche hazard, but mobile and free to travel unrestricted through the landscape".

2 Background

I'm still unclear why impact-based hazard mapping can't be used for backcountry recreation. Impact-based models can generate runouts without historical observations. Perhaps it's because impact-based models rely on low probability events, i.e., 10-300 year return periods?

7. We will change our explanation of this in the Background section, as the RAAMS model has been used in Switzerland to develop the Swiss terrain hazard maps (Harvey et al., 2018). Based on this, clearly an impact-based method can and has been used for backcountry recreation. But most of the time, impact-based methods are impractical for operations looking to incorporate ATES ratings. In lieu of running RAAMS on a landscape scale, which most operations cannot do, exposure-based methods are easier to implement and take advantage of localized terrain exposure knowledge.

L 71 Correct ATES Zoning model to ATES Zoning Model

8. Yes we will correct.

Section 3.1 This class 0 issue highlights the shortcomings of a model that doesn't use current conditions. No snow means no avalanche hazard, regardless of terrain. This limitation should be discussed.

9. Agreed, no snow means no avalanche hazard regardless of terrain. This same issue exists for dry snow climates, where there is often little or no snow but still an occasional snowpack and steep mountains (i.e.: leeward side of mountain ranges). To handle this, ATES relies strongly on the assessment of avalanche frequency (Section 4.2.5, Table 7).

Here are two examples:

1. A dry area that never receives a snowpack will have an avalanche frequency of *Never*, which defaults the ATES rating to Class 0. Compare this to a large, flat area with a deep snowpack climate where there is an avalanche frequency of *Never*, it will also be Class 0. Both of these locations will never see snow avalanches, thus they are Class 0.

2. An area of steep, complicated mountain terrain that is dry all summer may not have any summer avalanche hazard, but due to its snow climate may still have an avalanche frequency of 1:1 (events:years) because of the winter season, which will default the ATES rating to Class 3.

Class 0 is not a limitation; Class 0 is a feature of the ATES method and establishing it within ATES v.2 and in this manuscript is crucial. This factor was the primary motivation for updating the ATES system. It is fundamentally important (and difficult) to be able show where avalanches do not occur, and the fact that ATES v.1/04 did not have Class 0 was a significant limitation.

We see several areas where we can improve our manuscript to better explain this.

- We will modify the Background section to explain the limitation of not having Class 0 terrain in the original ATES, and the resulting conflict with ATES Zoning Model with does use Class 0.
- We will improve our description of avalanche frequency (Section 4.2.5) to ensure the interaction between snowpack and terrain are captured within ATES.
- We will improve our description on Line 160-161 of the ATES defaults around avalanche frequency to better show how much weight this carries.

Table 3 : 1:1 years? Please provide units. 10:1

10. Yes we will add units back in (we had them previously).

Section 4

The inputs to these tables are likely not available in many parts of the world, e.g., route options seems to require pre-mapped routes or trails. This could use some further explanation.

11. We will expand on Section 5.2 - Assessment methods (lines 427-453) to include a better explanation of how to these input tables can be assessed based upon data availability.

The inputs to these tables can be collected using various methods, both qualitative and quantitative. Certainly, some of the quantitative inputs (i.e.: DEM, forest cover) may not be available in many parts of the world and will require a generalized assessment, resulting in larger scale ratings. Other inputs (i.e.: frequency/magnitude) will mostly be subjective as avalanche frequency based on historical records or impact-based hazard mapping is not yet available on a landscape scale. Route Options and Exposure both require a location on the ground in order to assess, and this requires pre-mapped, or previously travelled routes or trails. Local knowledge of trails, backcountry routes or climbs is an essential input in lieu of pre-mapped routes. The system is intended to be used by field practitioners as well as desktop GIS specialists, ideally a team of both.

So, while all of these inputs would benefit from data that may not be available, the ATES system is designed to be subjective and allow field practitioners to undertake assessments based on

their observations and experience in the terrain, in lieu of hard data. Subjectivity is addressed on Lines 26 and 160, but we will improve this explanation in Section 5.2.

Table 4-6 are reprints of Campbell and Gould (2013). What's new here?

12. Addressed earlier in response #2.

214 –Unsupported slope needs clarification? With respect to what axis? Even a serac is supported in the slope normal direction.

13. The term "unsupported slope" is widely used by field practitioners to describe a slope on the vertical axis that rolls over suddenly and steeply at the top and offers minimal toe support at the base of the slope (i.e.: convex shape). In the mechanized ski industry, helicopter ski runs that are the least frequently open for skiing (i.e.: most hazardous) are characterized as having more unavoidable unsupported terrain shapes (Sterchi and Haegeli, 2019). Unsupported slopes are well described in Landrø et al., (2020). This contrasts with "supported slopes" that have a concave shape in the vertical axis, resulting a gradual transition to lower angle terrain and the perception of more toe support.

214 – "possible source". It's not a possible source. It is a source of tensile stress.

14. Agreed, we will change this.

Table 10 – representing an area with a point is an oxymoron

15. Representing an area on a map using a point is necessary at larger scales. Google Maps is the most obvious example of this, where areas, features, locations, etc. are displayed using a pin (point) on the map. As the map scale become smaller and more detailed, often these points will expand to show more detail. But at the larger scales, a point on the map is how you show an area.

5.1 Spatial scale

General comment, not the authors fault. The ISSW proceedings are not resolving on my browser without manually typing in. These proceedings need DOIs, not "persistent links" which don't resolve well and do not have changeable underlying links.  What happens when we are no longer using PHP, or arc.lib.montana.edu is no longer the base URL? Retrievability, especially for older works, is a major problem with grey lit citations. See my comments on preservation.

16. Agreed. This is one of the reasons we want to publish this manuscript and create a more robust, modern reference with a DOI for the ATES system.

Works cited:

Campbell, C. and Gould, B.: A proposed practical model for zoning with the Avalanche Terrain Exposure Scale, Proceedings of the 2013 International Snow Science Workshop, Grenoble, France, 385-391,

Dozier, J.: Revisiting topographic horizons in the era of big data and parallel computing, IEEE Geoscience and Remote Sensing Letters, 19, 8024605, 10.1109/LGRS.2021.3125278, 2022.

Dozier, J., Bruno, J., and Downey, P.: A faster solution to the horizon problem, Computers and Geosciences, 7, 145-151, 10.1016/0098-3004(81)90026-1, 1981.

Statham, G., McMahon, B., and Tomm, I. l. a. A., 2006.: The Avalanche Terrain Exposure Scale, Proceedings of the 2006 International Snow Science Workshop, Telluride, USA, 491-497,

Sturm, M. and Liston, G. E.: Revisiting the Global Seasonal Snow Classification: An Updated Dataset for Earth System Applications, Journal of Hydrometeorology, 22, 2917-2938, 10.1175/jhm-d-21-0070.1, 2021.

Sturm, M., Holmgren, J., and Liston, G. E.: A seasonal snow cover classification system for local to global applications, Journal of Climate, 8, 1261-1283, 10.1175/1520-0442(1995)008<1261:ASSCCS>2.0.CO;2, 1995.

Sykes, J., Toft, H., Haegeli, P., and Statham, G.: Automated Avalanche Terrain Exposure Scale (ATES) mapping – local validation and optimization in western Canada, Nat. Hazards Earth Syst. Sci., 24, 947-971, 10.5194/nhess-24-947-2024, 2024.

Toft, H. B., Sykes, J., Schauer, A., Hendrikx, J., and Hetland, A.: AutoATES v2.0: Automated Avalanche Terrain Exposure Scale mapping, Nat. Hazards Earth Syst. Sci., 24, 1779-1793, 10.5194/nhess-24-1779-2024, 2024.

Works cited:

Canadian Avalanche Association (CAA): Observation guidelines and recording standards for weather, snowpack and avalanches. Revelstoke, B.C., Canada, 109 pp, ISBN 0-9685856-3-9, 2016.

Haegeli, P., McCammon, I., Jamieson, B., Israelson, C. and Statham, G.: The Avaluator – A Canadian Rule-Based Avalanche Decision Support Tool for Amateur Recreationists, in: Proceedings International Snow Science Workshop, Telluride, USA, 1-6 October 2006, 254-263, https://arc.lib.montana.edu/snow-science/item.php?id=934, 2006.

Harvey, S., Schmudlach, G., Bühler, Y., Dürr, L., Stoffel, A. and Christen, M.: Avalanche Terrain Maps for Backcountry Skiing in Switzerland, in: Proceedings International Snow Science Workshop, Innsbruck, Austria, 7-12 October 2018, 1625–1631, https://arc.lib.montana.edu/snow-science/item.php?id=2833, 2018.

Jamieson, B. (ed.), 2018. Planning Methods for Assessing and Mitigating Snow Avalanche Risk, (contributions by Jamieson, B., Jones, A., Argue, C., Buhler, R., Campbell, C., Conlan, M., Gauthier, D., Gould, B., Johnson, G., Johnston, K., Jonsson, A., Sinickas, A., Statham, G., Stethem, C., Thumlert, S., Wilbur, C.). Canadian Avalanche Association, Revelstoke, BC, Canada.

Landrø, M., Pfuhl, G., Engeset, R., Jackson, M. and Hetland, A.:Avalanche decision-making frameworks: Classification and description of underlying factors, Cold Reg Sci Technol 169, https://doi.org/10.1016/j.coldregions.2019.102903, 2020.

Papathoma-Köhle, M., Neuhäuser, B., Ratzinger, K., Wenzel, H., and Dominey-Howes, D.: Elements at risk as a framework for assessing the vulnerability of communities to landslides, Nat. Hazards Earth Syst. Sci., 7, 765–779, https://doi.org/10.5194/nhess-7-765-2007, 2007.

Schmudlach, G and Köhler, J.: Automated Avalanche Risk Rating of Backcountry Ski Routes, in: Proceedings International Snow Science Workshop, Breckenridge, USA, 3-7 October 2016, 450-456, https://arc.lib.montana.edu/snow-science/item.php?id=2306, 2016.

Statham, G.: Avalanche Hazard, Danger and Risk – A Practical Explanation in: Proceedings International Snow Science Workshop, Whistler, BC, Canada, 21-25 September 2008, 224-227, https://arc.lib.montana.edu/snow-science/item.php?id=2939, 2008.

Statham, G. and Campbell, C.: The Avalanche Terrain Exposure Scale V.2, in: Proceedings International Snow Science Workshop, Bend, USA, 9-13 October 2023, 597-605, https://arc.lib.montana.edu/snow-science/item.php?id=2939, 2023.

Statham, G., Haegeli, P., Greene, E., Birkeland, K., Israelson, C., Tremper, B., Stethem, C., McMahon, B., White, B. and Kelly, J.: A conceptual model of avalanche hazard, Nat Hazards, 90, 663-691, https://doi.org/10.1007/s11069-017-3070-5, 2018.

Sterchi, R. and Haegeli, P.: A method of deriving operation-specific ski run classes for avalanche risk management decisions in mechanized skiing, Nat. Hazards Earth Syst. Sci., 19, 269–285, https://doi.org/10.5194/nhess-19-269-2019, 2019.

Sterlacchini, S. et al. (2014). Methods for the Characterization of the Vulnerability of Elements at Risk. In: Van Asch, T., Corominas, J., Greiving, S., Malet, JP., Sterlacchini, S. (eds) Mountain Risks: From Prediction to Management and Governance. Advances in Natural and Technological Hazards Research, vol 34. Springer, Dordrecht. https://doi.org/10.1007/978-94-007-6769-0_8

---

## Author Comment (AC2)

We would like to thank Stephan Harvey for his excellent review, which will result in important improvements to this manuscript. His experience with avalanche terrain, mapping and routefinding is clear and we sincerely appreciate his effort to improve our manuscript. Our responses are numbered and shown in red text below.

**ATES Paper Peer Review – Stephan Harvey – 06 July 2024**

General comments

The paper outlines the changes to the ATES terrain assessment based on the experience of the last 20 years. The main improvements are as follows:

a) additional classes 0 and 4,

b) exclusion of glaciers and

c) two types of definition (communication model and technical model).

The paper explains the ATES scale and how the classes are defined in a simple and comprehensible manner. The general outcome is not new.

The communication model provides a straightforward description of the scale, which helps less experienced people to recognise avalanche terrain. However, experience is required to understand the definition. E.g. Challenging: " With careful route finding, options exist to reduce or eliminate exposure".

The parameters "exposure" and "frequency magnitude" are subjective. I question whether "frequency magnitude" is a useful way to classify ATES. If the terrain characteristics are optimal for avalanches at a specific starting zone, then avalanches can occur if the snowpack is unstable. The question of frequency is more relevant when assessing objects with observed records. A remote starting zone with no records from past avalanches is not necessarily less complex than a starting zone were records are available.

1. We agree with the reviewer that if terrain characteristics are optimal for avalanches at a specific starting zone, then avalanches can occur if the snowpack is unstable. We also agree that the presence or absence or records does not change the complexity of the terrain; regardless of whether records exist or not, avalanche frequency is an important contributor to terrain severity and avalanche risk.

The point is that not all starting zones are created equally, and when the snow is unstable, certain terrain characteristics will produce avalanches more frequently in predictable locations than others. For example, a 40-degree, leeward slope that is reliably windloaded following every storm can be expected to release avalanches more frequently than a 25-degree, wind sheltered slope below treeline.

Certainly, avalanche frequency can be assessed with much higher reliability on avalanche paths with observed records. But only a small portion of backcountry avalanche paths have observed records. Many (most) known patterns of backcountry avalanche activity in specific locations result from informal, repeated field observations over many years. For example, the avalanche path "Frequent Flyer" in the backcountry of Rogers Pass, Canada, has no formal records, however it is known to release early in most storms and reliably produces several avalanches across the trail every winter.

This kind of information (formal or informal) is critical and has a direct influence the severity and hazardousness of the terrain. Terrain that produces avalanches frequently is more severe than terrain that produces avalanches infrequently. This is one of the most important drivers of terrain severity and is why ATES uses defaults to add weight to the avalanche frequency-magnitude categories.

Avalanche frequency is also crucial for how ATES handles terrain in dry snow climates, where there is usually little or no snow (no hazard), but occasionally a snowpack will develop and create an avalanche hazard (i.e.: every few years). In this case, even if the terrain is steep and complicated, if it is dry then there is no avalanche hazard, and an ATES rating must reflect that. This can be achieved by estimating avalanche frequency.

This is mostly a subjective category, with the rare exception of areas with long-term records. Table 7 is an example of commonly used categories of avalanche frequency in Canada. Despite similar tables being in widespread use in Canada, these appear to be mainly in consulting reports with no well-established references (that we could find). One of our objectives is to introduce such a table into the literature, as this method is widely used for terrain and avalanche hazard assessments (e.g.: CAA 2016; Jamieson 2018).

The authors say that the ATES rating is subjective and that some criteria require experience and local knowledge to be properly assessed. This leads to different assessments both for manually and automatic mapping.

2. Yes ATES v.2 is a mostly subjective terrain assessment method that is applied to trails, routes and climbs (guidebook style) as well as mapped. We have described this subjectivity in several spots in the manuscript (Lines 442-443 and 488-495) and encourage collaboration, field checking and peer review in order to check biases and improve consistency on the ratings. AutoATES removes these biases and provides consistent results, but AutoATES lacks the local knowledge of routes and avalanche frequency, so is crucially aided by expert knowledge at the smaller scales. Sykes et al., (2024) directly addresses the differences between human mapping and AutoATES and Figure 12 shows the validation of AutoATES against human mapping.

As some parameters of the technical model describe frequencies, this is only suitable for classifying an area or a route as a whole. It is not possible to assess individual cruxes or objects using some of the assessment criteria.  E.g. "many very large starting zones", "some open slopes > 35°", "options exist to avoid avalanche path".

3. We agree with the reviewer that some of the ATES v.2 criteria do not perform well at the terrain feature or slope scale, and that some of the assessment criteria require multiple slopes or avalanche paths to fully assess. Not all of the ATES criteria are available to be assessed for each situation, usually depending on the scale of the assessment. It was an oversight to not describe this, and we will ensure this is made clear in both Section 5.1 Spatial Scale and Section 5.2 Assessment Methods.

The possibilities of high-resolution terrain models and avalanche dynamics models to describe the runout, such as used in the Swiss avalanche terrain maps, are not mentioned. Furthermore, the different approach to classifying avalanche terrain is not mentioned, for example, "classified avalanche terrain, CAT" (e.g. in Introduction).

4. Agreed. This was an oversight as we have spent considerable time reviewing the work of Harvey et al., (2018). We will improve this manuscript's description of static versus dynamic hazard models as well as describe the Swiss CAT system. We will expand on this following Line 29 where the concept of the static model is introduced, and we will describe two key references regarding static and dynamic hazard maps (Harvey et al., 2018, Schmudlach and Köhler, 2016).

Specific comments

1 Introduction or 2 Background

Mention other automatic approaches, such as "Classified avalanche terrain, CAT" (Harvey et al., 2018), incorporating high-resolution digital terrain models, avalanche data and numerical avalanche dynamic model

5. Agreed. This was an oversight, and we will correct this in either the Introduction or the Background. We have spent considerable time reviewing the work of Harvey et al., (2018) and it was an oversight to not describe this and how static terrain mapping can enable dynamic hazard maps.

3 Principal changes to the Avalanche Terrain Exposure Scale

L80: has ATES Zoning Model been introduced before?

6. Yes, the ATES Zoning Model (Campbell and Gould, 2013) is introduced in the Background (Lines 70-78), but we will also add this to the Introduction following Line 30.

4 Avalanche Terrain Exposure Scale v.2

Tables 1 and 2 : The colours are not ideal:

7. We agree. Line 357-358 describes "further research is necessary to determine a colour standard that achieves the best balance of comprehension, base map visibility and CVD compliance". Engeset et al., (2022) is the only research into ATES colour choices, and their suggestion of black cross-hatching for Complex terrain will obscure the underlying base map.

We included Figure 9 to show a good example of alternative colour choices, and will improve our captioning to suggest the need for future work in this area.

Why is complex black and extreme red. Black is rather used for extreme in hazard rating.

8. ATES in Canada has always been black for Complex terrain, which was originally based upon the North American ski run difficulty system. European use of ATES subsequently changed this to red to align with the European ski run difficulty system. This is described in Lines 343-347. When Extreme (Class 4) terrain was added to ATES v.2, red was chosen to distinguish it from Complex terrain (black).

Black was originally a good choice for Complex terrain, as it was intuitive to skiers as a higher degree of terrain severity, but this was before ATES became a mapped product. Black is not ideal for mapping because it obscures the underlying base map information, and therefore adjusting its opacity is crucial to ensuring ATES maps are useable. This is described on Line 352.

Problem with red/green colour blindness.

9. This is a major challenge, as many warning systems in society (including avalanches) use both green and red. In order to communicate with a diverse audience, including those with Color Vision Deficiency (CVD), ATES uses a combination signal words, numbers and colours (Lines 333-335). This approach provides options for different ways to communicate with different people. To accommodate CVD issues, modern computers, websites and digital products can use colourblind filters which help with deuteranopia, protanopia, and tritanopia. Additionally, we consider the use of a map legend to be crucial. We will expand our description of this issue towards the end of Section 4.3.

L160-167: A bit short and confusing. Maybe pull up as matrix with ATES classes and 8 parameters defining 40 criterias. How are criterias ideally combines to rate ATES? I would like more guidance than in this respect.

10. We agree this is a confusing description of an important section and will revise Section 4.2 to introduce the ATES Technical Model (Table 3) more clearly. We will list the eight terrain factors to make it more obvious what is being assessed:

1. Exposure
2. Slope angle and forest density
3. Slope shape
4. Terrain traps
5. Frequency-magnitude
6. Starting zone size and density
7. Runout zone characteristics
8. Route options

In Section 5.2 Assessment Methods, we will then describe a four-step process to determine the rating, where each of the terrain factors is assessed against the five ATES categories (Class 0-4) and then combined in the following way:

1st  Assess each terrain factor independently and determine its rating level
2nd  Determine which ***default*** categories are met. This determines the minimum rating level
3rd  Compare each of the remaining terrain factors to the minimum rating level or higher
4th  For any categories higher than the minimum rating level, determine if this should outweigh the minimum rating level. This determines the ATES rating level.

We will also revise Section 5.2 Assessment methods, to describe the importance of considering the Communication models (Tables 1 and 2) in the assessment process. The Communication models are the message that the receiver gets, thus it is essential that the outcome of the assessment process described above (using the Technical model) aligns with the Communication model. Thus, the final step after determining ATES ratings with the Technical model, is to compare the ratings with the Communication model to ensure it is coherent.

Once the ATES rating has been determined using the Technical model and ensuring it aligns with the Communication model, peer review is a final step before publishing ratings.

Table 3:

Is "Frequency magnitude" necessary for backcountry ATES rating?

11. Yes, we have addressed this in reply #1.

Convoluted terrain leads to higher ATES rating than planar terrain (terrain shape). This contradicts with route options. In convoluted terrain there often are more options for less exposed route than in planar terrain. Furthermore planar terrain often leads to widespread crack propagation in unstable snowpacks and therefore to larger avalanches.

12. Excellent point and we agree with the reviewer that there are often more options for a less exposed route in convoluted terrain than in planar terrain; on the slope scale, a planar slope has less (no) options for routefinding, while a convoluted slope may provide more options for route selection and minimizing risk.

It is therefore the routefinding that is more complicated in convoluted terrain, not necessarily the avalanche exposure. Routefinding in planar terrain may be more straightforward, but this comes with a potentially higher degree of avalanche exposure as the reviewer describes, with more widespread crack propagation and less options to reduce risk through route selection.

This is a scale issue, and as described in response #3 above,  not all of the ATES v.2 criteria work well for smaller scale (terrain feature, slope) assessments; some criteria require multiple slopes to assess. We will make this clear in a revised manuscript. When looking at terrain at the larger scales, convoluted terrain shapes are more complex than isolated, single slope shapes that can

be navigated around to reduce risk. We propose to reword the description of slope shape accordingly:

| Slope shape | Straightforward, flat or undulating terrain | Straightforward undulating terrain | Mostly undulating with isolated slopes of planar, concave or convex shape | Convoluted, with multiple open slopes of intricate and varied terrain shapes | Intricate, often cliffy terrain with couloirs, spines and/or overhung by cornices |
|---|---|---|---|---|---|

Difficulty of defining "Exposure" and "Frequency magnitude". Also low frequence can be extreme terrain….

13. Defining exposure is not difficult when assessing a specific route or climb, in which case the exposure is known. For mapped terrain, when no specific route is assessed, we describe this as "potential exposure" on lines 175, 406 and 503. In this case, exposure cannot be defined until a route has been planned through the terrain at which point, the exposure is known and the ATES rating that affects that route can be determined.

Frequency-magnitude has been addressed in reply #1.

L220: Convoluted terrain: What do you want to say in this section?

14. We will revise this paragraph to explain the meaning more clearly.

Convoluted terrain presents a more spatially variable snowpack because the depth and distribution of snow is uneven as a result of wind redistribution through the topography. This uneven snowpack distribution increases spatial variability in the snowpack, resulting in more trigger points and increased uncertainty for snow stability evaluation (Schweizer at al., 2008). Zones of terrain that present mixed shapes of concave, convex and planar (Figure 4) usually present a snowpack with more trigger points than a zones with a smooth, evenly distributed snowpack where the depth and layer distribution is more predictable.

L224: What are propagation spots? Do you mean trigger spots?

15. Yes, we will change this to "trigger points".

L263: What about remote terrain without records? A slope with no records is not inherently less prone to avalanches.

16. Agreed, a slope with no records is not inherently less prone to avalanches. But regardless of whether records exist or not, a slope that avalanches more frequently than others is crucial information related to terrain severity. Determining this is a challenge, as records are rarely available for most backcountry terrain. Historically, avalanche experts have to make their assumptions often on a very weak basis (Maggioni and Gruber, 2003). Statistical methods exist, but these are usually inaccessible for smaller operations where the avalanche frequency of specific terrain and routes is known from experience with the terrain.

In lieu of records, common methods for assessing avalanche frequency include using local observations, knowledge, history and dendrochronology (Carrara, 1979).
* * *
References:

Canadian Avalanche Association (CAA): Observation guidelines and recording standards for weather, snowpack and avalanches. Revelstoke, B.C., Canada, 109 pp, ISBN 0-9685856-3-9, 2016.

Carrara, P.: The determination of snow avalanche frequency through tree-ring analysis and historical records at Ophir, Colorado. GSA Bulletin 1979; 90 (8): 773–780. https://doi.org/10.1130/0016-7606(1979)90<773:TDOSAF>2.0.CO;2, 1979.

Maggioni, M. and Gruber U.: The influence of topographic parameters on avalanche release dimension and frequency. Cold Reg Sci Technol 37, Issue 3, 407-419, https://doi.org/10.1016/S0165-232X(03)00080-6, 2003.

Jamieson, B. (ed.), 2018. Planning Methods for Assessing and Mitigating Snow Avalanche Risk, (contributions by Jamieson, B., Jones, A., Argue, C., Buhler, R., Campbell, C., Conlan, M., Gauthier, D., Gould, B., Johnson, G., Johnston, K., Jonsson, A., Sinickas, A., Statham, G., Stethem, C., Thumlert, S., Wilbur, C.). Canadian Avalanche Association, Revelstoke, BC, Canada.

---

## Author Comment (AC3)

We would like to thank Anonymous Referee #3 for an excellent and detailed review. It is obvious they have spent considerable time on this review and have provided us with many important revisions and suggested improvements. This will undoubtedly improve the quality of our manuscript and the ATES system, and we will endeavour to implement as many of these suggestions as possible into a revision.

Our responses are highlighted in red below:

**ATES Paper Peer Review – Anonymous Referee #3 – 10 July 2024**

The paper "The Avalanche Terrain Exposure Scale (ATES) v.2" by Statham and Campbell offers a valuable overview of the developments and applications of the ATES, focusing on the latest advancements. It presents ATES v.2 and the changes from ATES v.1, including applications to routes and areas. The paper effectively introduces ATES and its purpose, emphasizing the classification and communication of avalanche terrain exposure. The main updates from v.1 to v.2 are highlighted (5 classes instead of 3, removal of glaciation as a classification criterion) and the motivation behind the updates seems well argued. By providing background information on the technical model and the communication model(s) the manuscript is of interest to a wide target-audience ranging from avalanche professionals and educators to individuals working or recreating in potential avalanche terrain.

However, the main intention of the paper is partly unclear, which may cause potential readers to get lost between the general ATES review and the focus on developments specific to ATES v.2. While the historical evolution and widespread application of ATES are emphasized, it would be beneficial to also briefly discuss the similarities and differences to other (automated) avalanche terrain classification schemes (e.g. Harvey et al., 2018; Schmudlach and Köhler, 2016) and their reception by the target audience. To gain better insight into the ATES methodology, the systematic combination of terrain parameters to produce a rating is discussed in detail, but it would also be helpful to provide more guidance on how these parameters could be or are assessed.

1. We will expand the Background section of this manuscript to include descriptions of the work of Harvey et al., (2018) and Schmudlach and Köhler (2016) to develop other avalanche terrain classification schemes. It was on oversight not to include this in the original manuscript.

Based on this and similar feedback from Reviewer #2, we will improve our introduction of the ATES Technical Model (Section 4.2) to more clearly describe the assessment criteria and the process for combining them to determine the ATES rating.

We will list the eight terrain factors to make it more obvious what is being assessed:

1. Exposure
2. Slope angle and forest density
3. Slope shape
4. Terrain traps
5. Frequency-magnitude
6. Starting zone size and density
7. Runout zone characteristics
8. Route options

In section 5.2 Assessment Methods, we will then describe a four-step process to determine the rating, where each of the terrain factors is assessed against the five ATES categories (Class 0-4) and then combined in the following way:

1st  Assess each terrain factor independently and determine its rating level
2nd  Determine which *default* categories are met. This determines the minimum rating level
3rd  Compare each of the remaining terrain factors to the minimum rating level or higher
4th  For any categories higher than the minimum rating level, determine if this should outweigh the minimum rating level. This determines the ATES rating level.

We will also revise Section 5.2 Assessment methods, to describe the importance of considering the Communication models (Tables 1 and 2) in the assessment process. The Communication models are the message that the receiver gets, thus it is essential that the outcome of the assessment process described above (using the Technical model) aligns with the Communication model. Thus, the final step after determining ATES ratings with the Technical model, is to compare the ratings with the Communication model to ensure it is coherent.

Once the ATES rating has been determined using the Technical model and ensuring it aligns with the Communication model, field checking where possible and peer review are the final steps before publishing ratings.

The paper devotes considerable attention to the communication model and potential (risk management) applications, but the description of the updated technical model (thresholds, description) sometimes lacks consistency. One general point needing clarification in a revised version is the terminology. Specifically, the term "exposure" seems to be used inconsistently or with different meanings. On one hand, terrain exposure (potential vs. actual) towards avalanches is a crucial part of the terrain classification, while on the other hand, temporal exposure (of the element at risk) is used in terms of ATES as a risk management tool.

2. Thank you for highlighting this and we will review the entire paper for consistency in the use of the term "exposure". Specifically, in Section 4.2.1 Exposure, we will clarify the meaning of temporal exposure in the ATES model in the following way:

The assessment of ATES ratings considers temporal exposure in terms of "for how long" an element-at-risk is exposed, and the rating is dependent upon this. For example, being exposed to an avalanche path for 10 minutes presents a higher severity than being exposed to the same path for only 1 minute. The terminology *minimal, brief, intermittent, long, frequent* and *sustained* used in Tables 2 and 3 refers the length of time one should expect to be exposed to a piece of terrain.

The application (use) of ATES by the receiver considers temporal exposure in terms of "when" the different classes of terrain are within a risk threshold, and "when" they are not. This is independent of the element-at-risk and is a type of dynamic, avalanche risk assessment which requires combining the ATES rating (static) with avalanche hazard assessment (dynamic). For example, when the hazard is *Low*, then *Complex terrain* may be appropriate; alternatively, when the hazard is *High*, then *Complex terrain* may be inappropriate and *Simple terrain* the better choice. We will ensure that a revised manuscript addresses the dynamic nature of avalanche risk.

Additionally, the interplay between avalanche size and frequency could be discussed more in depth, including the limitations of a static representation of a dynamic problem (how the maps are connected to different avalanche hazard/problems/size scenarios).

3. We will address this in Section 4.2.5 Frequency and magnitude. Lines 280-283 touch on the interplay between avalanche size and frequency, and we will expand this section to explain the relationship between the two in more detail.

We agree that this manuscript should improve its description of static versus dynamic hazard models. This deserves an additional section either within, or following the Background section. Here we will describe both the limitations and the benefits of a static model and compare this with the dynamic hazard model. Importantly, static terrain maps are a foundation of dynamic hazard maps; an underlying DEM or even ATES map is necessary to combine with a dynamically modelled or remotely sensed snowpack to develop dynamic avalanche hazard maps. The whole point of ATES is to provide a static terrain exposure model with ordinal categories (Tables 1 and 2) that can be easily interpreted for public recreation and workplace avalanche safety.

Overall, the paper is timely, fits the target audience, and fills a gap in peer-reviewed literature on avalanche terrain classification schemes. Many recent scientific publications have been based on ideas developed by the authors over the past 20 years and as such a peer-reviewed reference to ATES is certainly of interest to the community. However, figures, captions, and referencing leave some room for improvement. Please refer to the specific line-by-line comments for more information.

Specific line by line comments

p.1 Abstract: The abstract is (nearly) identical to the one in the corresponding extended abstract in the ISSW23 proceedings. Please compare with the abstract in the corresponding ISSW

proceedings and revise accordingly. Two main questions appear in the abstract but also throughout the paper (see comments above):

l. 7-8: Are ATES ratings only independent of daily hazard conditions or also independent of the (temporal) exposure of the element at risk?

4. The use of ATES ratings as a tool for risk management is independent of the temporal exposure of the element-at-risk; a rating of Class 2 is valid regardless of whether the element-at-risk is exposed or not. However, the assessment of an ATES rating is dependent on the temporal exposure of the element-at-risk; the longer it takes to cross an avalanche path, the higher the exposure will be, and this should be reflected in the rating. We will clarify this in Section 4.2.1 Exposure.

l. 13-14: Is ATES actually risk management tool or is ATES a tool for risk management (comparable to what slope maps are for classical risk reduction methods)?

5. Subtle, but we appreciate the clarification in language and will change this accordingly.

p.2 l.19-23: Briefly explain the difference between hazard and risk, noting that hazard, vulnerability, and exposure are all key factors. Describe how these concepts are related to ATES and how exposure is defined within this context.

6. This is an important baseline, and we will describe the risk framework that ATES fits within (Statham, 2008; Statham et al., 2018) which will establish a reference point to define exposure within ATES.

p.2 l.30: Providing a more in-depth review of different applications of ATES, including manual and automated approaches, and applications in different regions would be interesting. Also an overview of different (spatial) application scales (location, size, etc.) could be of interest to the readers.

7. We will provide descriptions with references for interesting examples of manual versus automated approaches, different regional applications (.g.: Canada and Europe) and different recreational applications (skiing versus climbing).

p.2 l.40: Correct "Larson" to "Larsen." (generally check correct spelling and formatting of references in the manuscript) Yes

p.2 l.41: Insert a comma in "(Klassen, 2012)" and specify if this refers to ATES v.2 or ATES in general. Yes

p.2 l.48: Include both recreationists and professionals in the discussion.

8. We agree that terrain rating systems play an essential role for both recreationists and professionals and are directly relevant to both communities. But here we used the term "recreational" not to define the audience (recreationist or professional) but to define the

activity. Climbing, hiking, skiing and mountain biking are recreational activities, regardless of whether one is an amateur or a professional in their field and we intend this to mean that terrain rating systems are useful for recreational activities.

p.2 l.58-60: Expand on why quantitative methods become impractical in this context (due to the highly mobile element at risk). Clarify the impact and exposure-based approaches within ATES (see comments above on exposure).

9. We will improve our description of why quantitative methods become impractical (reviewer #1 has asked the same thing) when the element-at-risk is highly mobile. As described above, we will clarify temporal exposure in the context of both ATES, and exposure-based systems.

p.3 l.64: Clarify what is meant by "river ratings".

10. Penniman and Boisselle (1996) refer to the "river rating" system and show it in their Table 5 with no citation, but they mean the International Scale of River Difficulty (American Whitewater, 2024). We will change our manuscript to read ""… and modelled after river ratings, which describe the level of difficulty and the consequence of a rapid" and we will add a reference.

p.3 l.70: Reformulate the sentence for clarity. Yes.

p.3 l.76: Question the certainty that avalanches do not occur in class 0 terrain; suggest it may be very unlikely instead (compare table 2 "small sluffs").

11. This has been a challenge with defining Class 0 terrain, and we agree with the reviewer that there is a conflict with Table 2 where small sluffs and spindrift are possible in Class 0 (for ice climbing terrain). Any small snow slope can produce a small sluff, the key is that in Class 0 terrain it should be of no consequence.

We will change Line 76 to say: "the zoning model also introduced Class 0 (non-avalanche terrain), an essential rating level in any avalanche terrain classification system that shows where avalanches with consequence do not occur."

We will also modify Table 3, Class 0, Frequency-magnitude to say "***Never > size 1***".

Class 0 is presented as an optional terrain class, due to its need for a high degree of certainty. If the assessor remains uncertain, then Class 1 should be used. This is described on Lines 97-99.

p.8 l.173: Discuss how the ATES scenario (potential maximum avalanche size, expected average avalanche size, etc.) is implicitly considered when developing a spatial and temporal exposure rating. So any individual trying to come up with a spatial and temporal exposure rating for a certain location (for both ATESlinear and ATESspatial) must implicitly have some sort of "Avalanche Scenario" in mind (i.e. at least potential avalanche sizes, which are to some extent related to expected avalanche frequencies).

12. Maximum avalanche size and typical avalanche size are related directly to avalanche frequency-magnitude. Generally, it is expected that as the average frequency decreases down slope into the track and runout zone, the average magnitude increases (CAA 2016b). This is a key consideration when determining a rating, but doing this requires a route, or location (spatial exposure) to assess. Thus, these considerations apply to ATES linear, where the exposure is known but not directly to ATES spatial, with its "potential exposure". We address this in our revision of Section 4.2.5 Frequency and magnitude.

p.10 l. 191 ff. @Forest effects on avalanche formation:

The first argument you introduce in the discussion of forest effects is the mechanical anchoring of the snowcover in dense forest stands. However, this effect might be secondary to modification of snowpack structure by influencing wind re-distribution and micro-climatic conditions in forest stands. You also state that ATES uses a simplified model of the involved processes; maybe you can expand on which forest effects are included in your assessment and which are neglected (e.g. do you mainly consider forest effects on avalanche formation, or do you also consider potential forest effects on avalanche runout behavior?).

13. This is a good point as we have given a general explanation of forest effects but will be important to show the ATES forest density parameter considers primarily of the anchoring effects of the forest; snowpack modification is a secondary consideration. We will clarify this. ATES does not explicitly consider the forest effects on avalanche runout behaviour except that dendrochronology is an important input to avalanche frequency and thus is considered in this way.

Also a discussion of the extent of the spatial evaluation area might be of interest here, since average tree spacing might be substantially different when assessed over different spatial scales/extents (e.g. 10m pixel in GIS or a slope in a manual delineation).

14. Yes and in many cases the average tree spacing itself will determine the spatial extent of different zones for the reason described here. When average tree spacing is substantially different (in steep terrain) and results in a material change to the avalanche exposure, then the polygon boundaries (spatial extent) can represent these differences through zoning. See next comment #15.

How would you e.g. classify the slope depicted in Fig. 3 according to tables 5 and/or 6? Can you comment on how to assess average tree spacing in the scope of practical applications?

15. Figure 3 is part of a helicopter ski run located in the Purcell Mountains of British Columbia, Canada. For the entire area selected, the slope angle in the lower 50% is in the 20°- 30° range and the upper 50% is 30°- 45°, but nothing is steeper than 45° (Figure 3a). According to Table 6, this fits the criteria for Class 3 – Complex Terrain.

[Figure]

**Figure 3a.** Slope angle distribution across the area shown in Figure 3.

For the forest density distribution, the main open/gladed area in the middle is 120 stems/ha, slightly < 10 m spacing. The thicker forest around the area boundaries ranges from 209 stems/ha at the top, to 400-600 stems/ha on the northern flank, to 764 stems/ha at the bottom and southeastern corner (Figure 3b). According to Table 5, this is slightly below the threshold for Class 3 – Complex Terrain.

[Figure]

**Figure 3b.** Forest density distribution across the area shown in Figure 3.

The classification of this slope would depend on the objectives (Lines 390-393). Taken as one large area, the whole slope would be rated Class 3 – Complex Terrain, as the rating will default to the highest level within the area. However, smaller scale zoning would consider the different distributions of forest density and slope angle, resulting in some Class 1 and 2 terrain in the dense and gladed areas. There is perhaps even some Class 0 terrain in the SE corner of the area.

From a practical perspective, average tree spacing is done by estimating the distance between individual stems in various locations, and then applying this to the entire slope to estimate an

average value. The largest forest opening on the left side of the figure is 170 meters wide with slopes angles of 30°- 45°, so there is little protection from avalanches in this terrain. It would be possible to sneak through this terrain by following the contiguous strips of dense forest, however this is very close to the large open glades and would require previous, specific knowledge of the ski lines and a high level of skill to avoid the open slopes.

p.11 l.204: Provide more information on the reasons for combining ATES v.1 with the zoning model (such as automatic classification with more objective thresholds).

16. We agree that this is a gap in the manuscript and our revision will provide a better explanation of why ATES v.1/04 and the ATES Zoning model have been merged, however we will explain this in the Introduction section, approximately near to Line 37. The main reasons were that these two models had different thresholds, assessment criteria and descriptive terminology, and the Zoning Model's more objective approach to slope angle and forest density is an excellent baseline to support the subjective methods of the ATES Technical model.

p.11 l.211: Define "high spots" and "steep, unsupported slopes" for non-expert readers and explain their significance in relation to avalanche exposure (overhead hazard).

Yes.

p.12 l.220 ff.: Clarify the meaning of slope shape classes, as the definitions of classes between "flat" and "cliffy" are less clear. Discuss the slope shape factor's relevance to avalanche triggering and potential avalanche sizes and overhead hazards. All in all slope-shape is probably the least justified of the 8 ATES factors. The area reference remains rather unclear (while parts of a slope can be convex, the whole slope might be convoluted or intricate or cliffy?). The discussion of terrain shape is mainly linked to likelihood of avalanche triggering by limited additional load (e.g. skier-loading) and does not really consider factors such as potential avalanche sizes and overhead hazard?

17. Agreed, slope shape has been a difficult factor to justify and reference properly, but we consider the shape of the terrain to often be a (the) deciding factor when routefinding through avalanche terrain. This is a particularly difficult factor to justify in an objective manner, and its inclusion as an assessment factor is based upon the experience of professional ski and mountain guides.

In our response to Reviewer #2, we agreed that at the slope-scale there are usually more options for a less exposed route in convoluted terrain than in planar terrain; a planar slope has less options for routefinding, while a convoluted slope may provide more options for route selection and minimizing risk.

So it is the routefinding that is more complicated in convoluted terrain, not necessarily the avalanche exposure. Routefinding in planar terrain may be more straightforward, but this comes with a potentially higher degree of avalanche risk, with more widespread crack propagation and less options to reduce risk through route selection. ATES v.1/04 was designed to rate "routes"

(ATES linear), meaning that terrain that is more convoluted in shape with more challenging routefinding in avalanche terrain, receives a higher class of terrain rating.

This is also a scale issue, as some of the ATES v.2 criteria do not perform well at the terrain or slope scale, and some of the assessment criteria require multiple slopes or avalanche paths to fully assess. Assessing the slope shape criteria requires multiple slopes to fully assess properly. We will make this clear in a revised manuscript. When looking at terrain at the larger scales, convoluted terrain shapes are more complex than isolated, single slope shapes that can be navigated around to reduce risk. We propose to reword the description of slope shape accordingly:

| Slope shape | Straightforward, flat or undulating terrain | Straightforward undulating terrain | Mostly undulating with isolated slopes of planar, concave or convex shape | Convoluted, with multiple open slopes of intricate and varied terrain shapes | Intricate, often cliffy terrain with couloirs, spines and/or overhung by cornices |

However, we recognize that this factor is poorly supported with references and submit that it can be removed with few implications to the ATES method. We propose to keep the inclusion of slope shape in order to highlight its importance as a terrain factor, but we can highlight the lack of objective support and suggest these gaps in can be addressed in future avalanche terrain research.

p.13 @terrainTraps: Consider adding the Harvey et al. (2018) CAT, ATHM reference. Yes.

p.13 l.239, 245: Elaborate on what is meant by "harmless."

18. We will add a second sentence giving and example to provide context.

p.14 @Avalanche Frequency: Suggest a more nuanced formulation regarding the stability of avalanche frequency at a location, considering potential shifts due to climate change and the significance of the observation period.

19. Agreed. We will highlight potential changes to avalanche frequency due to shifts in climate patterns.

p 15. l. 274-276, Tab 8: Is e.g. typical impact pressure not rather a measure of intensity?

20. Yes, "Typical Impact Pressure" aligns with Jackson (2013) who defines intensity as "related to the effects at a specific location or area". However, Table 8 is taken directly from CAA (2016a), and we are reluctant to change an established definition that is the primary reference point for avalanche magnitude in Canada, even though we agree with the reviewer. We will pass this feedback to the CAA's Technical Committee for inclusion in an upcoming update.

p.16, section 4.2.7: Refine the definition of remote triggering. Use clear definitions for runout zones, distinguishing them from the track and relating them to international classification

schemes (e.g., zone of origin, transit, deposition, see De Quervain, R et al.: Avalanche atlas , Unesco, Paris, 1981 ). Yes.

p.16 l.302: Specify whether "The ATES Technical Model" refers to the old or the new model. Yes.

p.17 @route options: Clarify the meaning of "route options" when applied to ATES_spatial as opposed to ATES_linear.

21. Agree. This was an oversight and will be corrected.

p. 17 l. 320-321: Please check for consistency "Class 1" -> Class 1 Terrain, by checking for uniform usage throughout the text. Yes.

p.18: Consider providing specific color codes (RGB, hex) for clarity, as color descriptions alone can be vague. Yes.

p.19 l.345: Refer to the corresponding figure (Figure 11?) when discussing North American colors. Yes.

p.22 @actual vs. potential exposure: Clarify the difference between actual and potential exposure in the context of ATES, and how avalanche size scenarios play a role in identifying exposed segments. Additionally, (temporal) exposure in terms of avalanche risk may have a different meaning than (spatial) exposure to avalanche terrain (see comment above)?

22. Yes we will clarify the difference between actual and potential exposure and will link this to our improved description of temporal scale (see our response above).

p.22 @spatial scale: Discuss the importance of spatial scale and how well the eight ATES evaluation criteria can be assessed at different spatial scales (i.e. forest densitiy on a synoptic scale might be difficult, ...). Yes.

p.23 l.445: Provide a brief explanation of autoATES and related automated criteria and algorithms if it has not been referenced previously.

23. Yes, or we may move this statement (Lines 443-445) to the next Section 5.2.1 AutoATES.

p. 26 l.501: also refer to Larsen et al 2020 who developed autoATES v1 Yes

p.26 l.501: Discuss the limitation of automated algorithms for autoATES due to the 'subjective' selection of parameterization.

24. Yes we will add a sentence at the end of the paragraph describing this.

p.26 l.504-505: Potential & true (terrain or spatial) exposure: Please comment on the differences between the types of exposure and double check on the wording throughout the paper. Following your thought "because there is no route" further implies that without an element at risk there is no (temporal) exposure (see comment on "Is ATES a risk management tool or tool for risk management"?).

25. Yes, we will be improving our description of exposure (spatial, temporal, potential, actual) and will check for consistency throughout the manuscript.

p.26 l.506: Highlight the importance of being aware of resolution differences in maps and corresponding uncertainties. Yes.

p.27: Mention additional benefits of ATES compared to classical slope-angle-based terrain-choice strategies, particularly emphasizing the role of overhead hazard and terrain exposure to avalanches (one of the highlights of ATES, which appears to be underrated throughout the paper).

26. Yes, and we will look to improve our descriptions of overhead hazard throughout the manuscript.

Figures

General: Review all figures and their captions. Highlight key features in the images and provide appropriate scales for the maps.

Fig. 1: Describe the blue trails (ATES class 2?) and explain why the white trails are class 0 (due to flat and forested terrain?). Include data sources for trails and maps. Add an overview map for context, orientation (north arrow), and scale (scale bar). Yes.

Fig. 2, 4: Indicate if all displayed terrain is the same class and highlight accordingly.

27. Yes, we can do this also by including a dashed line to indicate the exact route that is referenced.

Fig. 3: Provide more context, such as slope angle, and show how an ATES v2.0 map would look in this area. Highlight different forest densities and slope angles.

28. Yes, this is good feedback as this image provides a good opportunity for this context and explanations. We do not think an ATES map will add value to this figure as the purpose of the image is to show the slope angle and forest parameters rather than the ATES rating itself.

Fig. 5: Highlight the zone of deposition and discuss terrain rating.

29. Yes we will annotate the figure to illustrate features in the terrain shapes.

Fig. 6, 7, 8: Highlight different zones and terrain ratings, particularly areas that might propagate into nearby starting zones. Yes.

Fig. 10: Include overview maps, orientation, and scale. Ensure 10b is not arbitrarily cut off and is easily interpretable without local knowledge; consider redesigning the figure. Yes.

References

Harvey, S., Schmudlach, G., Bühler, Y., Dürr, L., Stoffel, A., and Christen, M.: Avalanche Terrain Maps for Backcountry Skiing in Switzerland, in: Proceedings, International Snow Science Workshop, Innsbruck, Austria, 2018, pp. 1625 – 1631, Innsbruck, Austria, 2018.

Schmudlach, G. and Köhler, J.: Method for an automatized Avalanche Terrain Classification, in: Proceedings, International Snow Science Workshop, Breckenridge, Colorado, 2016, pp. 729 – 736, Breckenridge, Colorado, 2016.

De Quervain, R et al.: Avalanche atlas , Unesco, Paris, 1981
* * *
References

Canadian Avalanche Association (CAA): Observation guidelines and recording standards for weather, snowpack and avalanches. Revelstoke, B.C., Canada, 109 pp, ISBN 0-9685856-3-9, 2016a.

Canadian Avalanche Association (CAA): Technical Aspects of Snow Avalanche Risk Management - Resources and Guidelines for Avalanche Practitioners in Canada (C. Campbell, S. Conger, B. Gould, P. Haegeli, B. Jamieson, G. Statham Eds.). Revelstoke, BC, Canada, 117 pp, ISBN 978-1-926497-00-6, 2016b.

Jackson, L. E.: Frequency and Magnitude of Events. In Encyclopedia of Natural Hazards, Springer Netherlands, 359-363, https://doi.org/10.1007/978-1-4020-4399-4_147, 2013.

Penniman, D. and Boisselle, R.: Decision-making on Variable Risk Terrain, in: Proceedings International Snow Science Workshop, Banff, Canada, 6-11 October 1996, 67-72, https://arc.lib.montana.edu/snow-science/item.php?id=1406, 1996.

Statham, G.: Avalanche Hazard, Danger and Risk – A Practical Explanation, in: Proceedings International Snow Science Workshop, Whistler, BC, Canada, 21-27 September 2008, 224-227, https://arc.lib.montana.edu/snow-science/item.php?id=34, 2008.

Statham, G., Haegeli, P., Greene, E., Birkeland, K., Israelson, C., Tremper, B., Stethem, C., McMahon, B., White, B. and Kelly, J.: A conceptual model of avalanche hazard, Nat Hazards, 90, 663-691, https://doi.org/10.1007/s11069-017-3070-5, 2018.

Sykes, J., Haegeli, P., Atkins, R., Mair, P., and Bühler, Y.: Development of operational decision support tools for mechanized ski guiding using avalanche terrain modelling, GPS tracking, and machine learning, Nat. Hazards Earth Syst. Sci. Discuss. [preprint], https://doi.org/10.5194/nhess-2024-147, in review, 2024.

Walbridge, C. and Singleton, M.: Safety Code of American Whitewater. https://www.americanwhitewater.org/content/Wiki/safety:start?#vi, 2005, last accessed 20 August 2023.

---

## Author Response (AR1)

**NHESS ATES v.2 paper major revisions – TO DO**

This list of revisions includes formal feedback from the peer review process as well as feedback obtained outside of the formal peer review process. Each comment is followed by the initials of the source (reviewer), and a number that refers to our reply given during the interactive discussion.

For example, (AR3) refers to the third reply we gave our Anonymous Reviewer during the interactive discussion.

Revisions were received from the following sources:

(EB) = Edward Bair, (SH) = Stephan Harvey, (AR) = Anonymous reviewer, (HPM) = Hans-Peter Marshall (editor), (JF) = James Floyer, (BG) = Brian Gould, (BT) = Bruce Tremper, (CC) = Cam Campbell, (GS) = Grant Statham.

**NOTE:** Tables 1,2 and 3 were originally MS Word tables but have been replaced with .png images to meet NHESS formatting limitations on colour tables. The tables have better resolution and are preferred, but we will ask about this during the typesetting process if accepted. Sykes et al., (2024) reported the same issue, but was allowed to use the tables during typesetting. So this document has .png images as per NHESS request.
* * *
**Abstract** – Line 14 add that the Waterfall Ice climbing model is introduce with v2 (EB)

- Add the "objective of the paper" to the end of the abstract (EB5)
- Change Line 13 from "a risk management tool" to "a tool for risk management" (AR5) – did not change, too subtle and no significant impact.
- Abstract completely revised (AR early comment)

**Introduction** – After line 29 describe the differences between static and dynamic maps/rating systems (Schmudlach, Harvey) (SH4) (EB4).

- Introduce the Zoning Model after line 30 (SH6)
- Describe the objective of the paper to fill a gap in literature about terrain classification (describe the gap), get a baseline reference established in a peer reviewed journal with DOI reference (EB1) (EB16).
- Introduce the risk framework that ATES fits within, citing Statham (2008) and describing the interplay between hazard, exposure, vulnerability and risk. Best fit after Line 23 (AR6) (EB4).
- Line 30 provide a more in-depth review of different ATES applications (manual versus automated, different regions) and an overview of application scales (AR7).
- Line 37 describe why ATES v.1/04 and the Zoning Model were merged (AR16)
- Line 40 correct Laron to Larsen and insert comma into Klassen, 2012 (AR).

**Background** – At the end, after line 78, describe the Swiss maps, CAT rating system, Schmudlach ski touring guru (SH5) (AR1).

- Modify line 21 to read "…where the elements-at-risk such as skiers, climbers, snowmobilers or workers and exposed to avalanche hazard, but mobile and free to travel unrestricted through the landscape" (EB6).
- Lines 58-60 expand on why quantitative methods are impractical in this context (AR9).
- Line 57 revise to indicate the impact-based maps have been used RAMMS/FlowPy for backcountry applications but are not widely accessible to practitioners (EB7).
- Line 71 correct ATES Zoning Model from ATES Zoning model (EB8).
- Describe the limitations of ATES v.1 NOT having Class 0 and the conflict with the Zoning Model having it (EB9).
- Line 64 clarify river ratings "…modelled after river ratings, which describe the level of difficulty and the consequence of the rapid" and cite (American Whitewater, 2024) (AR10).
- Line 70 revise for clarity (AR)
- Change Line 76 to "the zoning model also introduced Class 0 (non-avalanche terrain), an essential rating level in any avalanche terrain classification system that shows where avalanches with consequence do not occur." (AR11).

**Section 4.1 ATES Communication models** – add more current NADS reference to line 143 (GS).

- Ensure the origin and overarching intent of ATES is public communication, everything else underneath is trying to define this (BT email).

**Section 4.2 ATES Technical Model** – introduce the TM criteria more clearly, describing in a list the eight terrain factors being assessed and linking them to the TM figure (Table 3) (SH10) (AR1).

- Improve the description of ATES defaults around avalanche frequency to show how they carry weight (the edit above may do this) (EB9).

**Section 4.2.1 Exposure** – Line 174 add that defining exposure is not difficult when assessing a route or climb, but exposure is difficult to define when no route exists (SH13).

- Clarify the meaning of temporal exposure showing the difference between *assessment* and *application* of ATES (AR2) (AR4).
- Review the entire paper for consistency in the use of the term exposure (AR2).
- Define actual versus potential exposure in the ATES context (AR22).

**Section 4.2.2 Slope angle and forest density** – Line 191 be more explicit in how ATES considers forest cover: primarily for "anchoring of the snowpack", while snowpack modification by the forest is a secondary consideration (AR13).

- Does ATES consider forest effects only for avalanche formation? Or also on runout behaviour? (AR13).
- Discuss the difference in spatial evaluation between a regional area (manual) and a 10m pixel in GIS. Discuss averaging and estimating versus measuring (AR14).
- Consider adding two figures from the review response and describe a practical application of slope angle and forest density towards a final rating (as shown in the response to reviewer, use this text) (AR15).

**Section 4.2.3 Slope shape** - check Tremper reference to convexities and revise to ensure this shows traditional thinking on convexities. Add references to ABS's and Avaluator on convexities (BT).

- Define "high spots" and "unsupported slopes" for non-expert readers and explain their significance in relation to avalanche exposure (overhead hazard) (AR)
- Describe the difference in route finding complexity versus avalanche complexity on convoluted terrain versus planar terrain and the limitation of the model at smaller scales (SH12).
- Line 220 revise this paragraph using the language of the SH peer review response (SH14).
- Change Line 224 to say "trigger spots" and review for correct meaning (SH14).
- Add a concluding statement (something like "convexity is therefore not used directly in ATES" or something similar to guide the reader about whether convexity is a direct factor or not in the rating) (HPM2)
- Line 214 improve the description of "unsupported slopes" consider referencing Landro (EB13).
- Line 214 remove the terms "possible source" and say "source" (EB14).
- Conclude that slope shape is poorly supported by research and describe a gap and the importance of this factor to practitioners (AR17).

**Section 4.2.4 Terrain traps** – reference Harvey et al. (2018) in this section (Swiss maps) (AR).

- Line 239 add a sentence giving an example of "an otherwise harmless avalanche" (AR18)

**Section 4.2.5 Frequency and magnitude** – referring to Line 263 describe the difference between "frequency" and "records" and that in the absence of records, frequency remains essential. It's the record keeping and the assessment of this parameter that is the challenge. Future options may improve our validation methods (satellites, remote sensing) (SH16).

- Ensure the limitations of frequency assessments are properly described.
- More nuance regarding determining the stability of avalanche frequency at location considering changes in climate and the limitations of the observation period (AR19).
- Describe other methods for assessing frequency beyond records and reference Carrara (1979) for tree rings (SH 16).

- Describe the assessment of F-M for a linear rating versus a zone, where exposure is known versus where exposure is potential. Consider describing the FM relationship with regards to downslope position in a path (AR12).

**Section 4.2.7 Runout zone characteristics** – refine the definition of remote triggering (AR)

- Clearly define the runout zone, distinguishing it from the track and relating them to international classification schemes (i.e. De Quervain : Avalanche Atlas) (AR) – this was reviewed and determined not necessary as the term "runout zone" is widely accepted as correct terminology, as opposed to the "zone of deposition" (DeQuervain, 1981) which is not widely used.
- Line 302 clarify ATES v.2 (AR).
- Line 320 – general comment check consistency of terminology throughout of Class 1 versus Class 1 Terrain (AR).

**Section 4.2.8 Route options** – clarify the meaning of route options when applied to ATES spatial versus ATES linear (AR21).

**Section 4.3 Signal words, colour and numbers** – improve the description of why black was chosen for Complex and red for Extreme using the language in the peer review response (SH8) somewhere associated with Line 352.

- Line 345 refer to a corresponding figure when describing colours (AR)
- Expand the description of the red/green colour blindness issue towards the end of Section 4.3 using the language in the peer review response (SH9).
- Describe colour choices and RBG codes referencing Sykes (2024) for the codes. See that peer review for a good description of language (AR). Coblis colour-blind simulator used.

**Section 4.4 Target audience** – add modern NADS reference to line 380 (GS)

**Section 5.1 Objectives and approach** – check section numbering, do we need this title or can we simply use 5.0 Application of ATES for the first part? (GS)

- Line 407 check actual versus potential exposure to ensure is aligns with the rewrite of section 4.2.1 Exposure (AR22).

**Section 5.1 Spatial scale** – describe how some ATES criteria do not perform well or cannot be used at "terrain feature" or "slope" scale because some criteria require multiple slopes to assess (SH3) (SH12).

- Section 5.1 is duplicated with a previous 5.1 Objectives and Approach. Check numbering (GS)
- Discuss the importance of spatial scale and how well the ATES criteria can be assessed at different scale (i.e.: forest density at regional scale is difficult) (AR).
- Line 425 expand on high resolution DEM versus lower resolution and its impact on terrain classification (GS).

**Section 5.2 Assessment methods** – describe how not all of the ATES criteria are available to be assessed for every situation. You use what you can get (SH3).

- Improve the description of how to manage subjectivity and combine with objective data where available (EB11).
- Describe how some data may not be available in some parts of world – use language from EB peer review response (EB11).
- Describe a four-step process to determine the rating where each of the terrain factors is assessed against the five ATES criteria and then combined (see peer review description) (SH10) (AR1).
- Describe the importance of aligning the TM ratings with the Communication Models (Table 1 and 2) as a final check (SH10) (AR1).
- Check line 444 (AutoATES) and consider expanding autoATES description or moving this statement to the next section on autoATES (AR23).

**6 Limitations** – line 501 include Larsen et al., 2020 for the original autoATES v.1 (AR).

- Line 501 discuss limitation of autoATES due to the "subjective" selection of parameterization (AR24) – did not do this, this paper is not about details of AutoATES.
- Line 504 – potential versus actual exposure – check to ensure this is used consistently throughout the paper and accurately describe its limitations (AR25).
- Line 506 – highlight importance of being aware of resolution in DEM maps and corresponding uncertainties.

**Conclusion** – describe additional benefits of ATES compared to classical slope angle based terrain choice strategies, emphasizing the role of overhead hazard (AR26)

- Review paper for opportunities to describe the role of overhead hazard in terrain exposure (AR26).

**Code and data availability** – create this new section to direct reader to the code bank for autoATES – check with John and use the same two banks he and Havard created (EB3) (HPM). Note this is not our code which is why it was not included originally, but two reviewers have asked for this, and we have received permission from Larsen and Sykes to publish its location here.
* * *
**Figure 1** – describe all trails (currently blue) and explain why white trails are class 0 (flat, forest, etc.) (AR)

- Include data source for trails and maps (parks Canada data) (AR)
- Include overview map for context, N arrow and scale bar (AR)

**Figure 2** – annotate figure to show the route through the couloir that is rated. Describe why its rated Extreme. Consider adding some terrain measurements (AR27).

**Figure 3 –** provide more context (slope angle), show how an ATES v.2 map would classify this terrain (AR28).

- Consider adding the two figures (slope angle/forest distributions) to help explain (GS).

**Figure 4** – show routes through the terrain, describe it as Complex and why (AR27)

**Figure 5** – highlight the deposition zone and discuss the terrain rating and why (AR29) describe planar slope feeding directly into a terrain trap.

**Figures 6,7,8** – highlight different zones and terrain ratings, particularly areas that might propagate into nearby starting zones (AR). Good place to show frequencies.

**Figure 9** – improve the caption to describe the need for additional research into colour choices (SH7) (EB2)

- Include new figure obtained directly from Huber et al and check Christophe's email comments on this figure to add to caption (GS).

**Figure 10** – Include overview maps, orientation and scale. Ensure 10b is not arbitrarily cut off and is easily interpretable without local knowledge; consider redesigning the figure (AR).
* * *
**Tables 1, 2 and 3** – change to figures to accommodate NHESS limitation on colour tables, then revise all figure numbers and check all references.

**Table 3** – Revise language in Slope Shape row using language from SH peer review response (SH12) (AR17).

- Revise slope angle and forest density according to James Floyer suggestion (JF)
- Replace units (avalanches:years) on frequency section (EB10).
- Class 0 frequency-magnitude change to "**Never > size 1**" (AR11).

**Table 4** – clarify meaning of "large proportions" and try to use a % value (HPM1). It was ultimately removed from this table.

- Check slope angle terminology/definitions to ensure they align with Comm/Tech models

**Table 5** – cite as "modified from Campbell and Gould (2013)." (EB)

**Table 6** – cite as "modified from Campbell and Gould (2013)." (EB)

**Table 7** – Change "Rare" to "Extremely low" (BG/CC)

**Table 9** – Add RGB and Hex codes to the table following/citing Sykes (2024) (AR)

Grant Statham and Cam Campbell – November 27, 2024

---

## Author Response (AR2)

**Nhess-2024-89**

**The Avalanche Terrain Exposure Scale (ATES) v.2**

Author's response to Editor and Executive Editor comments – February 11, 2025

Dear Editors,

We are pleased to see you have accepted our manuscript pending technical corrections. Please find below our explanations of the corrections/changes we have made to the FINAL manuscript.

1.  We have removed **Section 8 Code Availability** and added the same information into **Section 5.3 AutoATES** (lines 652-654). This is the more appropriate section for referencing this code, which is not our code (Sykes et al., 2024; Toft, et al., 2024) and is referenced here using a doi link to directs readers to the AutoATES code repository.

2.  With regards to the Editor's comment *"It would be very helpful if even just a few paragraphs could be added to the README.md file explaining how to install and run the code."* We have contacted John Sykes who maintains this code bank and requested that he add this explanation to the code. Because this is not our code and we do not maintain this repository, we cannot make these changes ourselves.

3.  We understand the request to NOT have coloured tables as these cannot be published in the final revised version. Thus, we have included Tables 1, 2 and 3 as .png files (images) but continue to refer to them as Tables (even though they are technically now an image). We are prepared to change this if necessary, but Sykes et al. (2024) encountered the same issue with a similar table last year and it was published in this way. See Table 1 in https://doi.org/10.5194/nhess-24-947-2024. Sykes described this issue as being solved through the typesetting process. We are prepared to modify if necessary.

We hope this satisfies your request for technical corrections, and please advise if you require anything further.

Sincerely,

Grant Statham and Cam Campbell